# One-hot Generalized Linear Model for Switching Brain State Discovery

**Chengrui Li[1], Soon Ho Kim[2], Chris Rodgers[3], Hannah Choi[2] & Anqi Wu[1]**

**[1] School of Computational Science & Engineering**
Georgia Institute of Technology, Atlanta, GA 30305, USA
`{cnlichengrui,anqiwu}@gatech.edu`

**[2] School of Mathematics**
Georgia Institute of Technology, Atlanta, GA 30305, USA
`{soonhokim,hannahch}@gatech.edu`

**[3] Department of Neurosurgery, School of Medicine**
Emory University, Atlanta, GA 30322, USA
`christopher.rodgers@emory.edu`

## Abstract

Exposing meaningful and interpretable neural interactions is critical to understanding neural circuits. Inferred neural interactions from neural signals primarily reflect functional connectivity. In a long experiment, subject animals may experience different stages defined by the experiment, stimuli, or behavioral states, and hence functional connectivity can change over time. To model dynamically changing functional connectivity, prior work employs state-switching generalized linear models with hidden Markov models (i.e., HMM-GLMs). However, we argue they lack biological plausibility, as functional connectivities are shaped and confined by the underlying anatomical connectome. Here, we propose two novel prior-informed state-switching GLMs, called Gaussian HMM-GLM (Gaussian prior) and one-hot HMM-GLM (Gumbel-Softmax one-hot prior). We show that the learned prior should capture the state-invariant interaction, shedding light on the underlying anatomical connectome and revealing more likely physical neuron interactions. The state-dependent interaction modeled by each GLM offers traceability to capture functional variations across multiple brain states. Our methods effectively recover true interaction structures in simulated data, achieve the highest predictive likelihood, and enhance the interpretability of interaction patterns and hidden states when applied to real neural data. The code is available at `https://github.com/JerrySoybean/onehot-hmmglm`.

## 1 Introduction

Unveiling meaningful and interpretable neural interaction structures is vital for comprehending neural circuits. Extensive research has investigated these interactions using statistical and information-theoretic methods like cross-correlogram (Jia et al., 2022), mutual information (Houghton, 2019), Granger causality (Granger, 1969), transfer entropy (Schreiber, 2000), generalized linear methods on top of dynamical systems (Linderman et al., 2016; 2017; Glaser et al., 2020), and Hawkes process (Li et al., 2022). Typically, the inferred neural interaction reflects dynamic functional connectivity, influenced by variations in neural activity. Direct observation or inference of the anatomical connectome, encompassing axons, dendrites, and synapses that establish neural communication, is usually not feasible. Moreover, unlike anatomical connectome, which remains relatively stable over a period of time, function connectivity varies with behavioral states and on much faster time scales. Functional connectivities of neurons, therefore, reflect dynamic modes of computation shaped by task and sensory inputs. Existing experimental evidence suggests that neural systems can exhibit diverse firing patterns associated with different sensory, perceptual, and behavioral states (Sherman, 2001; Haider et al., 2007; Anderson et al., 2000; Sanchez-Vives & McCormick, 2000; Escola et al., 2011; Wang et al., 2024).

To capture such time-varying functional connectivities in multi-state systems, prior studies explored state-switching generalized linear models (GLMs) with hidden Markov models (HMMs), referred

to as HMM-GLMs (Escola et al., 2011; Nadagouda & Davenport, 2021; Zhou et al., 2021; Morariu-Patrichi & Pakkanen, 2022). These models introduce a discrete hidden variable representing the state of each time point, with each state equipped with its own GLM to capture neural interactions. However, we argue that such methods are not biologically plausible enough, since an interaction between a pair of neurons inferred from neural signals can reflect not only functional connectivity but also anatomical connectome or synaptic connectivity. There exists experimental evidence manifesting degrees of correlations between functional and anatomical networks (Genç et al., 2016; Siegle et al., 2021). It is thus plausible to assume that functional connectivity is dynamically modulated by brain states while also being shaped and confined by the underlying anatomical connectome.

Incorporating these more biologically plausible assumptions, we introduce a novel approach for capturing time-varying functional connectivity in multi-state neural systems using an HMM-GLM framework. Unlike previous HMM-GLM methods that assume complete independence among GLMs in different states, we introduce a learnable prior for all states, constraining the search space for the interaction weight of each GLM derived from neural activity. We first provide a solution using a shared Gaussian prior over the weight matrices of GLMs for all states, denoted as Gaussian HMM-GLM. However, this Gaussian prior is relatively naive and does not explicitly connect functional connectivities to the anatomical connectome. Accordingly, we provide a second solution that decomposes each GLM's weight matrix into an adjacency matrix and a strength matrix, with the adjacency matrix modeled by a one-hot encoding mechanism. The prior is then imposed solely on the adjacency, not the entire weight matrix. We argue that the regulated adjacency matrices, guided by their prior, shed light on the underlying anatomical connectome, revealing more likely physical interactions between neurons. Meanwhile, less restricted strength matrices offer flexibility to capture functional variations across multiple brain states. Our experimental results demonstrate that when compared to alternatives, one-hot HMM-GLM accurately recovers true interaction structures in simulated data and achieves the highest predictive likelihood on test spike trains from two real neural datasets. Moreover, the uncovered interaction structures and hidden states are more interpretable compared with alternatives in real neural datasets.

## 2 METHOD

**Classic GLM**: Denote a spike train data as $\boldsymbol{X} \in \mathbb{N}^{T \times N}$ recorded from $N$ neurons across $T$ time bins, $x_{t,n}$ as the number of spikes generated by the $n$-th neuron in the $t$-th time bin, and $\boldsymbol{x}_t \in \mathbb{R}^{N \times 1}$ as the vector of spikes for all neurons at time $t$. When provided with $\boldsymbol{X}$, a classic GLM, with pre-defined basis functions, predicts the firing rates of the $n$-th neuron at the time bin $t$ as

$$f_{t,n} = \sigma \left( b_n + \sum_{n'=1}^{N} w_{n \leftarrow n'} \cdot \left( \sum_{k=1}^{K} x_{t-k,n'} \phi_k \right) \right), \quad \text{with spike } x_{t,n} \sim \text{Poisson}(f_{t,n}), \quad (1)$$

where $\sigma : \mathbb{R} \to \mathbb{R}_+$ is a non-linear function (e.g., Softplus); $b_n$ is the background intensity of the $n$-th neuron; $w_{n \leftarrow n'}$ is the influence weight from the $n'$-th neuron to the $n$-th neuron whose matrix form is $\boldsymbol{W} \in \mathbb{R}^{N \times N}$; $\boldsymbol{\phi} \in \mathbb{R}_+^K$ is the basis function summarizing history spikes from $t - K$ to $t - 1$. The GLM finds the optimal $\boldsymbol{W}$ by maximizing the Poisson log-likelihood of the observed spikes.

**HMM-GLM (HG)**: First, we extend the GLM with a hidden Markov model (HMM). We assume there exist $S$ states underlying the functional connectivities of neural activity. For each time $t$, we introduce a discrete latent variable $z_t \in \{1, \dots, S\}$, whose transition probability is $p(z_{t+1}|z_t) = \pi_{z_t, z_{t+1}}$ with a matrix form $\boldsymbol{\Pi} \in [0,1]^{S \times S}$. Given a latent state $z_t$, the weight matrix at time $t$ is selected as $\boldsymbol{W}_{z_t} \in \mathbb{R}^{n \times n}$ from $\{\boldsymbol{W}_s\}_{s=1}^{S}$. Then the emission model is $p(x_{t,n}|z_t, \boldsymbol{x}_1, \dots, \boldsymbol{x}_{t-1}) = \text{Poisson}(f_{t,n})$:

$$f_{t,n} = \sigma \left( b_n + \sum_{n'=1}^{N} w_{z_t, n \leftarrow n'} \cdot \left( \sum_{k=1}^{K} x_{t-k,n'} \phi_k \right) \right). \quad (2)$$

Note that the traditional HMM framework assumes that the emission probability distributions, similar to the transition probability distributions, are time-homogeneous, i.e., the emission model does not depend on any previous observations. Here we relax the assumption by introducing the dependence over the spike history, similar to the previous HMM-GLMs (Escola et al., 2011). Next, we will introduce two variants of HG, in which a single shared global prior is imposed over all $\boldsymbol{W}_s$ in different states.

**Gaussian HMM-GLM (GHG)**: To impose the assumption that functional connectivities across different states should share some common structure, a straightforward approach is imposing a Gaussian prior $\mathcal{N}(w_{0, n \leftarrow n'}, \sigma^2)$ on the weight $w_{s, n \leftarrow n'}$ with hyperparameter $\sigma^2$, $\forall s \in \{1, \dots, S\}$. An

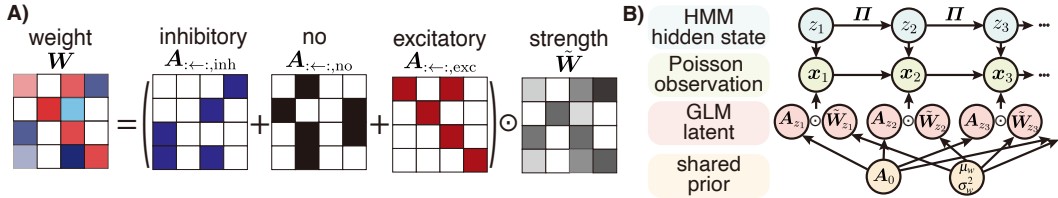

Figure 1: A) A descriptive schematic of the weight matrix decomposition. B) The graphical model of the one-hot HMM-GLM.

HG with such a Gaussian prior is referred to as Gaussian HMM-GLM (GHG), and the shared common information of state-dependent weights $\boldsymbol{W}_s$ is stored in $\boldsymbol{W}_0$.

**One-hot HMM-GLM (OHG)**: One-hot HMM-GLM (OHG) decomposes the weight matrices $\boldsymbol{W}_s$ in Eq. 2 into a discrete adjacency matrix and a positive-valued strength matrix, i.e.,

$$w_{s,n\leftarrow n'} = [(-1)a_{s,n\leftarrow n',\text{inh}} + (+1)a_{s,n\leftarrow n',\text{exc}}] \cdot \tilde{w}_{s,n\leftarrow n'}, \quad \forall s \in \{1,\dots,S\}. \tag{3}$$

$\tilde{w}_{s,n\leftarrow n'} \in \mathbb{R}_+$ is the strength of the weight in state $s$. We define $\boldsymbol{a}_{s,n\leftarrow n'} = [a_{s,n\leftarrow n',\text{inh}}, a_{s,n\leftarrow n',\text{no}}, a_{s,n\leftarrow n',\text{exc}}] \in \Delta^2$ to be the adjacency (weight type) from neuron $n'$ to neuron $n$ in state $s$ corresponding to {inhibitory, no connection, excitatory}. $\boldsymbol{a}_{s,n\leftarrow n'}$ is a soft one-hot encoding vector over a simplex $\Delta^2 := \{\boldsymbol{a} \in [0,1]^3 | \sum_{i=1}^3 a_i = 1\}$, so its scaler representation $a_{s,n\leftarrow n'} = (-1)a_{s,n\leftarrow n',\text{inh}} + (+1)a_{s,n\leftarrow n',\text{exc}}$ should be close to either $-1$ or $0$ or $+1$. The matrix and tensor forms are denoted as $\tilde{\boldsymbol{W}}_s \in \mathbb{R}_+^{N\times N}$ and $\boldsymbol{A}_s \in [0,1]^{N\times N\times 3}$ respectively, $\forall s \in \{1,\dots,S\}$. Fig. 1A shows a schematic of the one-hot decomposition.

To impose the assumption that functional connectivities across different states should share some common structure informing us about the underlying anatomical connectome, we impose a Gumbel-Softmax prior over $\boldsymbol{a}_{s,n\leftarrow n'}$, i.e., $\boldsymbol{a}_{s,n\leftarrow n'} \sim \text{Gumbel-Softmax}(\boldsymbol{a}_{0,n\leftarrow n'},\tau)$, $\forall s \in \{1,\dots,S\}$, written out as

$$a_{s,n\leftarrow n',\text{type}} = \frac{\exp\left[(\ln a_{0,n\leftarrow n',\text{type}} + g_{s,n\leftarrow n',\text{type}})/\tau\right]}{\sum_{\text{type}'\in\{\text{inh,no,exc}\}} \exp\left[(\ln a_{0,n\leftarrow n',\text{type}'} + g_{s,n\leftarrow n',\text{type}'})/\tau\right]}, \forall\,\text{type} \in \{\text{inh, no, exc}\} \tag{4}$$

where $g_{s,n\leftarrow n',\text{type}} \overset{\text{i.i.d.}}{\sim} \text{Gumbel}(0,1)$. In practice, we can sample $g$ by sampling $u$ from $\text{Uniform}(0,1)$ and computing $g = -\ln(-\ln(u))$. $\tau > 0$ is a temperature hyperparameter forcing $\boldsymbol{a}_{s,n\leftarrow n'}$ to be a soft one-hot representation of the weight type. For each $\boldsymbol{a}_{s,n\leftarrow n'}$, only one of the three types is significantly hotter than the other two, representing the type of the connection. The tensor form of $\boldsymbol{a}_{0,n\leftarrow n'}$ is denoted as $\boldsymbol{A}_0 \in \mathbb{R}^{N\times N\times 3}$, which is a free-parameter matrix imposing the biological structure similarity over different states. Consequently, if the synaptic (anatomical) connection type is excitatory, its functional connection type is likely to be excitatory; and vice versa. The log density of the Gumbel-Softmax distribution Jang et al. (2016); Maddison et al. (2016) is:

$$\ln p(\boldsymbol{a}_{s,n\leftarrow n'}|\boldsymbol{a}_{0,n\leftarrow n'}) = \ln 2 + 2\tau - 2\ln\left(\sum_{\text{type}\in\{\text{inh,no,exc}\}} \frac{a_{0,n\leftarrow n',\text{type}}}{(a_{s,n\leftarrow n',\text{type}})^\tau}\right)$$
$$+ \sum_{\text{type}\in\{\text{inh,no,exc}\}} \left(\ln a_{0,n\leftarrow n',\text{type}} - (\tau+1)\ln(a_{s,n\leftarrow n',\text{type}})\right). \tag{5}$$

By introducing a Gumbel-Softmax prior over the connection matrix $\boldsymbol{A}_s$, we turn the parameter $\boldsymbol{A}_s$ into a latent variable. We can also assume the strength $\tilde{\boldsymbol{W}}_s$ and the background intensity $b_n$ are random variables from some prior distributions. We put a Gaussian prior over the log of $\tilde{\boldsymbol{W}}_s$ to ensure its positivity and a Gaussian prior over $b_n$, finally obtaining the generative model of OHG:

$$z_{t+1}|z_t \sim \text{Categorical}(\pi_{z_t,1},\dots,\pi_{z_t,S}), \quad \forall t \in \{1,\dots,T\}$$
$$\boldsymbol{a}_{s,n\leftarrow n'} \sim \text{Gumbel-Softmax}(\boldsymbol{a}_{0,n\leftarrow n'},\tau), \quad \forall s \in \{1,\dots,S\}, \forall n,n' \in \{1,\dots,N\}$$
$$\ln \tilde{w}_{s,n\leftarrow n'} \sim \mathcal{N}(\mu_w,\sigma_w^2), \quad \forall s \in \{1,\dots,S\}, \forall n,n' \in \{1,\dots,N\} \tag{6}$$
$$b_n \sim \mathcal{N}(\mu_b,\sigma_b^2), \quad \forall n \in \{1,\dots,N\}$$
$$x_{t,n} \sim \text{Poisson}(f_{t,n}(\boldsymbol{x}_1,\dots,\boldsymbol{x}_{t-1},\boldsymbol{A}_{z_t},\tilde{\boldsymbol{W}}_{z_t},b_n)), \forall t \in \{1,\dots,T\}, \forall n,n' \in \{1,\dots,N\}.$$

A schematic diagram representing this generative model is shown in Fig. 1B.

**Relationships between GHG and OHG**: GHG is similar to OHG in the sense that they both assume that the state-dependent weights $\boldsymbol{W}_s$ share some common information ($\boldsymbol{A}_0$ for OHG and $\boldsymbol{W}_0$

for GHG). The main difference is that GHG does not differentiate the adjacency from the strength (Eq. 3). Therefore, the shared $\boldsymbol{W}_0$ incorporates both. While in OHG, thanks to the decomposition, $\boldsymbol{A}_0$ only imposes similarity over the adjacency, not the strength. The regulated adjacency matrices $\boldsymbol{A}_s$ with their prior $\boldsymbol{A}_0$ should inform us about the underlying anatomical connectome. The less restricted strength matrices $\tilde{\boldsymbol{W}}_s$ provide us with sufficient flexibility to capture functional variations across multiple brain states. Compared with OHG, GHG serves as an intermediate model with a straightforward prior directly on the weights representing shared global connectivity but without the one-hot decomposition. In the experimental evaluation section, we will show that a biologically plausible constraint like $\boldsymbol{A}_0$ in OHG is critical to obtaining meaningful inference and learning results.

## 3 INFERENCE

The generative model has four latent variables $\{z_t, \boldsymbol{A}_s, \ln \tilde{\boldsymbol{W}}_s, b_n\}$. It requires a complex fully Bayesian inference approach to infer all the latent variables, which is usually very time-consuming and highly computationally intensive. We provide a Baum-Welch algorithm to solve the inference problem. In our Baum-Welch, we derive the posterior of $z_t$ in the E-step, and do maximum likelihood estimation for all other latent variables given the estimated posterior distribution of $z_t$ in the M-step, i.e., we jointly optimize model parameters and latent variables (except $z_t$) in the M-step. The rationale is that the calculation of the posterior for $z_t$ is straightforward via forward-backward message passing, while the calculation of the posterior for $\boldsymbol{A}_s$ is very challenging and has no closed-form expression. We can certainly resort to a variational distribution to approximate the posterior for $\boldsymbol{A}_s$. However, since the prior of $\boldsymbol{A}_s$ is a Gumbel-Softmax distribution, it is unclear what parametric density function we should choose to serve as the approximated posterior distribution. Given these challenges, we only do the E-step for $z_t$ with forward-backward message passing. In the M-step, we optimize the model parameters $\{\boldsymbol{\Pi}, \boldsymbol{A}_0\}$ with $\{\boldsymbol{A}_s, \ln \tilde{\boldsymbol{W}}_s, b_n\}$, denoted as $\theta$ altogether. The hyperparemeter set is $\zeta = \{\mu_w, \sigma_w^2, \mu_b, \sigma_b^2, \tau\}$, which is pre-defined, detailed later. We also pre-define the basis function $\boldsymbol{\phi} \in \mathbb{R}_+^K$.

First, we infer the hidden state given $\theta^{\text{old}}$ with the forward-backward algorithm (E-step). In this step, we will omit $\theta^{\text{old}}$ for simplicity. We define $\gamma_{z_t}(t) := p(z_t|\boldsymbol{X}; \theta^{\text{old}})$, $\xi_{z_{t-1}, z_t}(t) := p(z_{t-1}, z_t|\boldsymbol{X}; \theta^{\text{old}})$, and define $\alpha_{z_t}(t) := p(\boldsymbol{x}_1, \ldots, \boldsymbol{x}_t, z_t)$, $\beta_{z_t}(t) := p(z_{t+1}, \ldots, z_T|\boldsymbol{x}_1, \ldots, \boldsymbol{x}_t, z_t)$. Then, we can obtain the relationship $\gamma_{z_t}(t) = \frac{\alpha_{z_t}(t)\beta_{z_t}(t)}{p(\boldsymbol{X})}$, $\xi_{z_{t-1}, z_t}(t) = \frac{\beta_{z_t}(t)p(\boldsymbol{x}_t|\boldsymbol{x}_1, \ldots, \boldsymbol{x}_{t-1}, z_t)\alpha_{z_{t-1}}(t-1)p(z_t|z_{t-1})}{p(\boldsymbol{X})}$. $\alpha_{z_t}(t)$ and $\beta_{z_t}(t)$ can be computed iteratively as

$$\begin{cases} \alpha_{z_t}(t) = p(\boldsymbol{x}_t|\boldsymbol{x}_1, \ldots, \boldsymbol{x}_{t-1}, z_t) \sum_{z_{t-1}=1}^{S} \alpha_{z_{t-1}}(t)p(z_t|z_{t-1}), & \alpha_{z_1}(1) = p(z_1)p(\boldsymbol{x}_1|z_1) \\ \beta_{z_t}(t) = \sum_{z_{t+1}=1}^{S} \beta_{z_{t+1}}(t+1)p(\boldsymbol{x}_{t+1}|\boldsymbol{x}_1, \ldots, \boldsymbol{x}_t, z_{t+1})p(z_{t+1}|z_t), & \beta_{z_T}(T) = 1 \end{cases}$$

resulting in $p(\boldsymbol{X}) = \sum_{z_T=1}^{S} \alpha_{z_T}(T)$. With this inferred posterior for $\boldsymbol{z}$, we can update $\theta$ in the M-step by maximizing

$$\begin{aligned} Q(\theta, \theta^{\text{old}}) =& \mathbb{E}_{p(\boldsymbol{z}|\boldsymbol{X}; \theta^{\text{old}})} \ln p(\boldsymbol{X}, \boldsymbol{z}; \theta) = \sum_{\boldsymbol{z}} p(\boldsymbol{z}|\boldsymbol{X}; \theta^{\text{old}}) \ln p(\boldsymbol{X}, \boldsymbol{z}; \theta) \\ =& \sum_{z_1=1}^{S} \gamma_{z_1}(1) \ln p(z_1; \theta) + \sum_{t=2}^{T} \sum_{z_{t-1}=1}^{S} \sum_{z_t=1}^{S} \xi_{z_{t-1}, z_t}(t) \ln p(z_t|z_{t-1}; \theta) \\ &+ \sum_{t=1}^{T} \sum_{z_t=1}^{S} \gamma_{z_t}(t) \ln p(\boldsymbol{x}_t|\boldsymbol{x}_1, \ldots, \boldsymbol{x}_{t-1}, z_t; \theta). \end{aligned} \quad (7)$$

More details about the inference can be found in Appendix A.1.1.

There are several key hyperparameters in $\zeta$ requiring pre-defining before inference. (1) Gumbel-Softmax temperature $\tau$: It is common to choose the temperature $\tau$ in Gumbel-Softmax from $[0.1, 1]$. If $\tau$ is too large, the relaxation will be too soft; if $\tau$ is too small, numerical issues could arise. In our model, $\tau$ is used to force the soft one-hot close to one corner of the simplex, so we tried $\tau \in \{0.1, 0.2, 0.5\}$, and found that the result of the one-hot HMM-GLM is not sensitive to $\tau$ in this range. Given that the selection of $\tau$ is insensitive to different datasets, we fix $\tau = 0.2$, which is a common moderate choice. (2) Generative hyperparameters $\{\mu_w, \sigma_w^2, \mu_b, \sigma_b^2\}$: we chose $\mu_w = -5, \sigma_w = 2$ and $\mu_b = 0, \sigma_b = 2$ since this set provides noninformative priors for the weight strength and the background intensity in GLMs, and hence the inference is insensitive to different datasets.

## 4 EXPERIMENTAL EVALUATION

**Models for comparison.**    We will compare our methods and state-of-the-art baseline methods on one simulated data and two real neural datasets:
- **GLM** (Pillow et al., 2008): The most original model for discovering neural interactions, without the multiple-state assumption.
- **HMM Corr** (Engel et al., 2016): An HMM for discovering state switches from spike train data. Since this method cannot find neural connectivities but only the latent states, we use a correlation-based method, i.e., cross-correlogram (CCG) to find the connectivity in each inferred state.
- **HMM Bern** (Ashwood et al., 2022): Similar to the HMM Corr, but uses the Bernoulli rather than Poisson distribution to model the spike count in each time bin.
- **HG** (Escola et al., 2011): The naive HMM-GLM (HG), which is the only existing model that both infers latent states and learns neural connectivities.
- **GHG** (our method): We abbreviate Gaussian HMM-GLM as GHG.
- **OHG** (our method): We abbreviate one-hot HMM-GLM as OHG.
- **HG-L1** and **GHG-L1**: Given that the one-hot mechanism implicitly imposes sparsity on the weight matrix, concerns may arise regarding whether the imposition of sparsity solely accounts for OHG's superiority. To address this, we add two more models: one by adding an L1 penalty to the weight of HG (HG-L1), and another to the weight of GHG (GHG-L1). The L1 penalty coefficients are determined through validation.

**Metrics.**    We use the following metrics to report performances from different models:
- **LL**. The log-likelihood on the test set. A better model should have a stronger ability to predict future spiking events. Note that this is the only metric that can be used on real-world datasets, since there are usually no true states and neural connectivities available for real-world datasets.
- **State accuracy**. The average accuracy of the inferred states across all time bins. This is only applicable to the simulated dataset where we know the true hidden states.
- **Weight error**. The error of the learned weight matrices in all states. Note that there is no weight error for HMM Corr and HMM Bern. Since their learned weights are from CCG, the weights cannot be compared with the weights in the GLM model. This is only applicable to the simulated dataset.
- **Adjacency accuracy**. The balanced accuracy of the learned adjacency matrices in all states. This is only applicable to the simulated dataset. For models without adjacency matrices explicitly modeled, we use

$$\boldsymbol{a}_{s,n\leftarrow n'} = \begin{cases} \left(0, 1 - \frac{w_{s,n\leftarrow n'}}{\max_{s,n,n'} w_{s,n\leftarrow n'}}, \frac{w_{s,n\leftarrow n'}}{\max_{s,n,n'} w_{s,n\leftarrow n'}}\right), & w_{s,n\leftarrow n'} \geqslant 0 \\ \left(\frac{w_{s,n\leftarrow n'}}{\min_{s,n,n'} w_{s,n\leftarrow n'}}, 1 - \frac{w_{s,n\leftarrow n'}}{\min_{s,n,n'} w_{s,n\leftarrow n'}}, 0\right), & w_{s,n\leftarrow n'} < 0 \end{cases} \tag{8}$$

to obtain the adjacency matrix from the learned weight matrix. We choose Eq. 8 since it is an automatic way with a reasonable rationale. We can also use a pre-defined threshold to obtain the adjacency matrix, but the accuracy of the connection matrices is very sensitive to the thresholding technique (see Appendix A.2). In real neural data analysis, when we don't have the ground-truth adjacency matrices, we cannot even use such an accuracy metric to select the optimal threshold value. This demonstrates that the explicit adjacency matrices from the OHG provide a succinct expression requiring no pre-defined thresholds but rendering satisfactory estimation.
- **Adjacency prior accuracy**. This is only applicable to the simulated dataset. Except for OHG, the adjacency prior is obtained by first averaging the weight matrices across all states and then applying the averaged weight to Eq. 8.

### 4.1 APPLICATION TO SIMULATED DATA

**Dataset.**    We first compare different models on a 5-state-20-neuron synthetic dataset with 10 independent trials. For each trial, we generate 20 spike sequences of length $T = 5000$. Each spike sequence is generated from the generative model in Eq. 6, with $\pi_{s,s'} = 0.005 + 0.975 \cdot \mathbb{1}[s = s']$, $\tau = 0$, $\mu_w = -5, \sigma_w^2 = 1.5$, and $\mu_b = 0, \sigma_b^2 = 0.0008$. We sample $\boldsymbol{a}_{0,n'\leftarrow n}$ from Dirichlet$(0.1, 0.8, 0.1)$, $\forall n, n' \in \{1, \ldots, 20\}$. Note that when $\tau = 0$, all $\boldsymbol{A}_s$ are hard one-hot encodings i.i.d. sampled from $\boldsymbol{A}_0$. This is equivalent to sample $a_{s,n\leftarrow n'}$ from a categorical distribution, i.e., $a_{s,n\leftarrow n'} \sim \text{Categorical}(a_{0,n\leftarrow n'})$, $\forall s \in \{1, \ldots, 5\}$, $\forall n, n' \in \{1, \ldots, 20\}$. This introduces some mismatching generative procedures compared with Eq. 6. For each trial, we train different models on the first 10 sequences and test on the remaining 10 sequences.

The quantitative results in Tab. 1 show that OHG is the best in terms of all five metrics. GHGs are the second best since GHGs impose shared global prior on the weights. However, they are still worse than OHG, validating that the one-hot component accounts for the better performance of OHG.

Table 1: The quantitative results in terms of 5 metrics on the synthetic dataset.

| method | LL ↑ | state acc ↑ | weight error ↓ | adj acc ↑ | adj prior acc ↑ |
|--------|------|-------------|----------------|-----------|-----------------|
| GLM | -8.43(±0.18) | nan(±nan) | 24.71(±0.19) | 43.12(±0.46) | 44.81(±0.61) |
| HMM Corr | -22.53(±0.64) | 42.84(±1.47) | nan(±nan) | 34.04(±0.12) | 15.45(±2.49) |
| HMM Bern | -5.68(±0.23) | 87.95(±0.93) | nan(±nan) | 36.25(±0.25) | 40.70(±1.53) |
| HG | -5.49(±0.58) | 37.73(±2.80) | 109.67(±2.63) | 34.17(±0.08) | 40.91(±0.48) |
| HG-L1 | 9.14(±0.18) | 91.60(±0.96) | 23.14(±0.08) | 37.47(±0.18) | 48.44(±0.57) |
| GHG | 8.58(±0.19) | 91.80(±0.92) | 21.54(±0.15) | 42.53(±0.22) | 48.93(±0.54) |
| GHG-L1 | 9.77(±0.20) | 92.08(±0.89) | 14.16(±0.07) | 41.08(±0.22) | 46.98(±0.60) |
| OHG | **14.64**(±0.23) | **92.75**(±0.87) | **10.99**(±0.21) | **73.90**(±0.52) | **80.60**(±0.59) |

Figure 2: Visualization of weight $W_2$ (top row) and adjacency $A_2$ (middle row) corresponding to state 2 ($S = 5$ in total), and the adjacency prior $A_0$ (bottom row) for all models learned from one trial of the synthetic dataset.

Next, we analyze the neural connectivities learned by different models (Fig. 2). Although there are $S = 5$ different states, one-state GLM only captures an "average" estimation across the 5 states. For HMM Corr and HMM Bern, the learning procedure is decoupled into two steps, inferring hidden states and estimating the neural connectivity in each inferred state. Although the inferred hidden states from HMM Bern are acceptable, the estimated adjacency in each state and the adjacency prior are still bad. For HG, the poor performance is mainly from an incorrect estimation of the transition matrix, which leads to a bad inference of the hidden state sequence (Fig. 7 in Appendix 7) and hence results in a wrong weight and adjacency estimation. Comparing HG with GHG and OHG, we conclude that a constraint (i.e., a prior) on different states is necessary for a stable result, since the shared information between different states can help prevent the inferred states and the weights in different states from falling into extremes or bad local optima. Adding an L1 penalty could suppress some of the noisy weights but is still not helpful for estimating the adjacency in each state and the shared adjacency prior, as L1 does not enhance discrimination between a weak and no connection. The main difference between GHG and OHG is their weight and adjacency estimations. GHG still has many noisy non-zero weights. With the one-hot setting in OHG, the sparsity of the network is easily learned, and connections with zero interactions are successfully suppressed, which leads to a lower weight error and better adjacency accuracy (the weights, adjacencies, and the adjacency prior learned by OHG match the true the best in Fig. 2).

## 4.2 APPLICATIONS TO ELECTROPHYSIOLOGY DATA

### 4.2.1 PREFRONTAL CORTEX DURING A CONTINGENCY TASK

**Dataset.** We first apply different models to a prefrontal cortex (PFC-6) dataset (Peyrache et al., 2018; 2009)[1]. Neural spike trains were collected while a rat learned a behavioral contingency task. During recording, the animal performed a trial for about 4 secs and then took a short break for about 24 secs. The spike train data used for learning and testing is segmented from the long session. Each sequence starts from 5 seconds before a behavior start and lasts for 10 seconds after the start. Hence, each sequence corresponds to a behavioral trial. We use $\frac{2}{3}$ of the neural sequences as the training set and the remaining $\frac{1}{3}$ as the test set. The neural spikes are binned into 750 time bins with bin size = 20 ms. Since we do not know the true number of hidden states, we try $S \in \{2, 3, 4, 5\}$.

---

[1] https://crcns.org/data-sets/pfc/pfc-6

Table 2: The log-likelihood on the test set for different models and different numbers of states of the PFC-6 dataset. The result from the one-state GLM is -36.35($\pm$0.00).

| method | 2 states | 3 states | 4 states | 5 states |
|---|---|---|---|---|
| HMM Corr | -37.11($\pm$0.00) | -36.60($\pm$0.00) | -36.53($\pm$0.00) | -36.68($\pm$0.00) |
| HMM Bern | -36.89($\pm$0.00) | -36.57($\pm$0.00) | -36.38($\pm$0.00) | -36.38($\pm$0.00) |
| HG | -37.30($\pm$0.05) | -37.61($\pm$0.17) | -37.22($\pm$0.14) | -36.98($\pm$0.19) |
| HG-L1 | -36.91($\pm$0.01) | -36.90($\pm$0.02) | -36.73($\pm$0.09) | -36.63($\pm$0.13) |
| GHG | -37.17($\pm$0.00) | -37.11($\pm$0.01) | -37.12($\pm$0.00) | -37.11($\pm$0.00) |
| GHG-L1 | -36.94($\pm$0.00) | -36.88($\pm$0.00) | -36.83($\pm$0.00) | -36.77($\pm$0.00) |
| OHG | **-35.92**($\pm$0.02) | **-35.79**($\pm$0.02) | **-35.77**($\pm$0.03) | **-35.71**($\pm$0.03) |

Figure 3: Visualization of weight $W_4$ (top row) and adjacency $A_4$ (middle row) corresponding to state 4 ($S = 4$ in total), and the adjacency prior $A_0$ (bottom row) for all models learned from the PFC-6 dataset.

Tab. 2 shows that the test log-likelihoods of OHG with all different numbers of states are consistently better than others. Fig. 3 shows an example of the weights and adjacencies estimated by different models. For HG, the learned weight matrices are pretty dense and noisy, resulting in a bad test log-likelihood. For GHG, the weights are less dense but still noisy. Adding L1 penalties to HG and GHG helps reduce some noisy weight entries, but does not help discriminate between a weak connection and no connection. Using OHG, we can get a much clearer strength-adjacency decomposition and also obtain an adjacency prior. The global restriction provided by the adjacency prior shapes the functional connectivities as the anatomical connectome does, which improves OHG's test log-likelihood. Note that GLM actually achieves a reasonably good result, only worse than OHG. This indicates that in such real-world scenarios, functional connectivities in different states indeed share a global static structure (may reflect the anatomical connectome), outweighing the functional differences between different states and hence should be taken into account. For $S > 5$, the performance does not increase significantly and becomes flat (Fig. 10 in Appendix A.4).

Although there is no ground truth of hidden states, we can integrate the behavioral data to analyze the inferred hidden states from different models. Pick 4 states as an example. In Fig. 4, we plot the hidden state prediction of one incorrect trial (Fig. 4A) and one correct trial (Fig. 4B). We also plot the corresponding rat movement on the right-hand side and color it according to the inferred states from OHG. As previously observed, HG continues to yield a state prediction characterized by significant noise and limited interpretability. Although the number of hidden states is set as $S = 4$, GHG only infers two effective hidden states. The transition from state 4 to state 3 typically happens when the rat turns back at the wrong target location. However, OHG is able to find four explainable effective hidden states. Before each trial, the rat goes back to the root of the Y-shaped maze (starting point), corresponding to state 4. Then the rat turns around at the starting point and goes forward to the turning point of the Y-shaped maze, corresponding to state 3. After making the decision, the rat enters into state 2 in one arm of the Y-shaped maze, to reach the destination. If the rat goes to the correct target location, it gets a reward at the target and will stay in state 4 for a long while. But if the rat goes to the incorrect target location, there is no reward and the rat will go back immediately, corresponding to state 1. This state interpretation is reflected in the colored rat trajectory in Fig. 4. Note that these state patterns are not from cherry-picking. We do observe similar state transitions

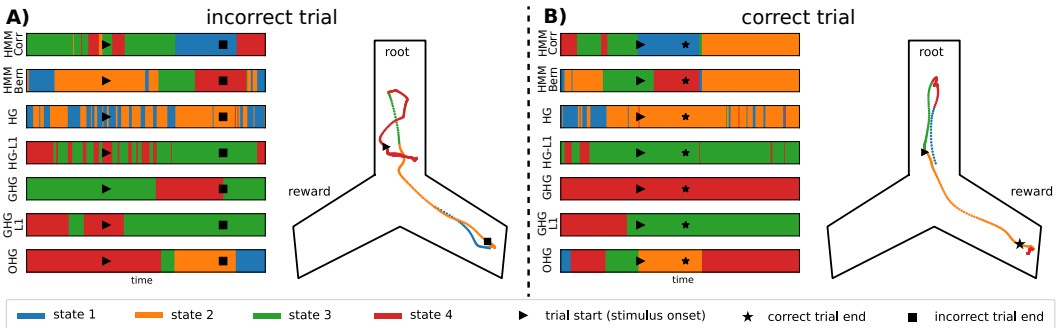

Figure 4: Inferred hidden states of an incorrect trial (A) and a correct trial (B) from various models including HG, GHG, and OHG. The rat trajectory on the right-hand side of each one is colored according to the hidden states inferred from OHG.

among other more correct and incorrect trials, which can be checked and validated in Fig. 8 in Appendix A.4. We also perform an analysis of task information decoding from the inferred hidden states (Fig. 9 in Appendix A.4) to further validate the rationality this interpretation.

### 4.2.2 BARREL CORTEX DURING WHISKING

**Dataset.** We next apply different models to electrode recordings of the somatosensory (barrel) cortex in mice during a shape discrimination task (Rodgers et al., 2021; Rodgers, 2022; Nogueira et al., 2023) (Fig. 5A). Mice were trained to discriminate concave from convex shapes using only their whiskers. In particular, the mice are required to actively whisk in order to make contact with the object; a high-speed video of whisker motion was collected, allowing analysis of the active movement of the whiskers to sense the environment. Here we use 27 sessions from 5 different mice. The number of recorded neurons varies from 10 to 44 across sessions. Six seconds from each trial is included in the analysis, and spike trains are discretized with a time bin of 3 ms. The first 30 trials are used in the analysis of each session, of which 10 randomly selected trials form the test set when evaluating the test log-likelihood, and the remaining 20 trials are used for training the model.

Given that we do not have good knowledge about the behavioral states, we try different numbers of hidden states for the barrel cortex data, i.e., $S = \{2, 3, 4, 5\}$. The log-likelihoods of the models fit to the barrel cortex dataset show similar trends to the PFC-6 dataset. OHG consistently has the highest log-likelihood, and GHG generally exhibits greater log-likelihood compared to other baseline models across different numbers of hidden states (See Fig. 5B).

Fig. 5C shows the whisker positions, contacts, and inferred hidden states of each model. We select the case of $S = 2$ hidden states here for visualization. While the log-likelihood of OHG increases as $S$ increases to 5, for $S > 2$, there are many sessions with rarely occupied states, and the distinction between different states becomes subtle. Results for 3-5 states are shown in Appendix A.5. When two states are assumed, it is typically observed that one of the states inferred by GHG and OHG coincides with active whisking events during which contacts occurred, while the states inferred from HG switch very frequently.

While GHG and OHG correlated with whisking events similarly, the durations of the inferred states are different (Fig. 5C). OHG infers a stable state with a duration over 1 s that persists over whisking cycles, while the inferred states from GHG switch rapidly with short durations ($< 0.1$ s). OHG thus better captures sustained whisking cycles (Deschênes et al. (2012); Rodgers et al. (2021)). Fig. 5D further shows the weight and adjacency matrices, and the adjacency prior estimated by each model for the same session shown in Fig. 5C. As in the PFC-6 dataset, we observe that only OHG learns sparse and clear weight matrices, while the ones learned by HG and GHG are denser and noisier.

We further test the idea that the states inferred from GHG and OHG are related to the active whisking events statistically. We compute the frequency with which whisker contacts are initiated in each state, and perform a $\chi^2$ test against the expected frequencies if no relation between the states and contacts is assumed. Among 11 sessions where all three models result in inferred state frequencies that are not completely skewed (the least frequent state was inferred in at least $5\%$ of the time steps), the null hypothesis is rejected ($p < 0.001$) in 6 sessions (54%) for HG and in 8 sessions (73%) for both GHG and OHG. Furthermore, across all sessions, we compute the sum of all elements in the weight matrix $W$ of the state associated with whisker contacts and that of the other state. When

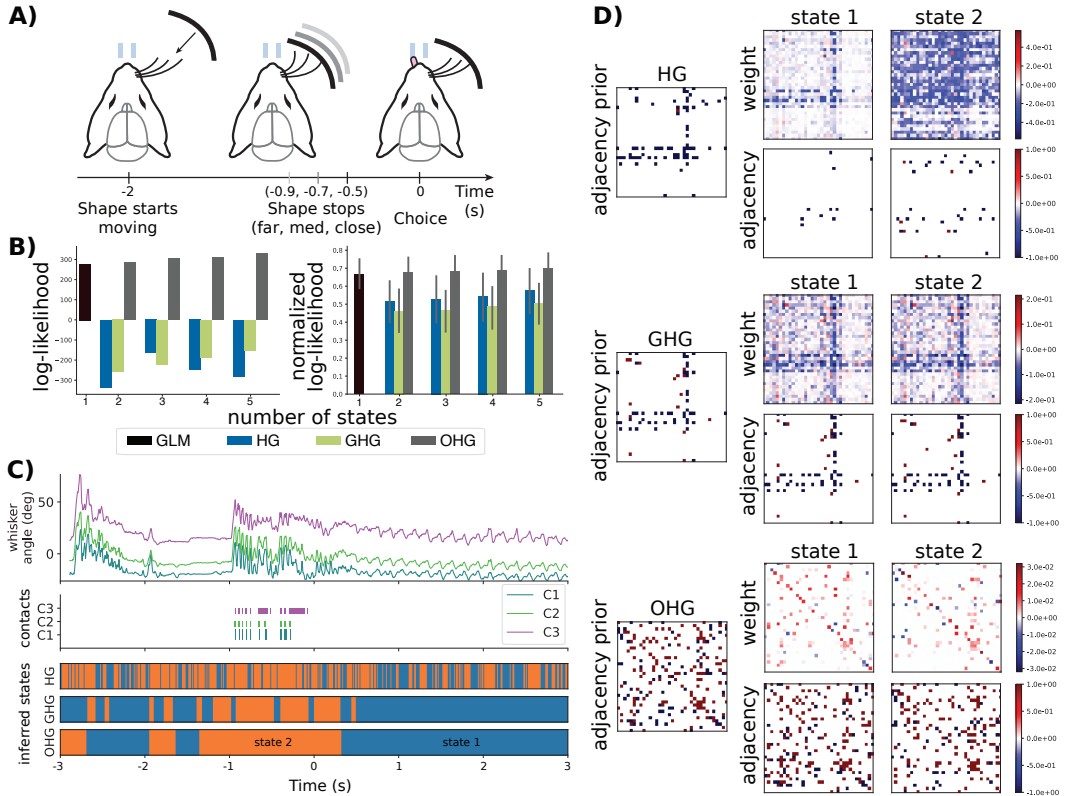

Figure 5: A) Experimental setup for the whisking task (adapted from Nogueira et al. (2023)). B) Test (normalized) log-likelihoods from different models. Error bars are not shown for raw log-likelihood (left) due to extremely high session-by-session variation. To account for this, the normalized log-likelihood is also shown (right). C) An example trial of whisker positions (top panel), whisker contacts with the object (middle), and the inferred states (bottom). $t = 0$ s is the time at which the response window is opened (after which the licking direction of the mouse is considered as its decision), and the stimulus is presented at approximately $t = -1$ s. D) Weights, adjacencies, and adjacency priors of the three models in the example session shown in C.

comparing the distribution of total weight between whisking and non-whisking states, OHG results in a significant increase of the weights during whisking states ($p = 0.008$, two-sided Wilcoxon rank-sum test), while HG and GHG do not ($p > 0.1$). This suggests that OHG is capable of detecting shifts in functional interaction tied to switching behavioral states.

## 5 CONCLUSION

We develop a novel one-hot HMM-GLM (OHG) to estimate time-varying functional connectivity in multi-state neural systems. The newly proposed OHG decomposes the traditional weight matrix in GLMs into a discrete adjacency matrix representing the connection type and a positive-valued strength matrix. When building OHG, we place a common Gumbel-Softmax prior over the adjacency matrices for all states, enforcing the adjacency matrices to learn shared information. We argue that the regulated adjacency matrices with their shared prior should inform us about underlying anatomical connectome and thus uncover the "more likely" physical interactions between neurons. The less restricted strength matrices are allowed to change freely without a shared prior across different states, and hence can provide us with sufficient flexibility to capture functional variations across multiple brain states. We argue that OHG is more biologically plausible given the aforementioned benefits. Gaussian HMM-GLM (GHG) serves as an intermediate model with shared prior directly on weight matrices without (one-hot) strength-adjacency decomposition, confirming that such a decomposition is critical for the success of multi-state inference. The experiments show that when compared with alternatives, OHG gets better connectivities and hidden states. It not only accurately recovers the true connectivities for simulated data but also achieves the best predictive likelihood on test spike trains for a PFC dataset and a barrel cortex dataset. The uncovered connectivities and hidden state sequence from OHG are more interpretable for these real neural datasets.

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

# A APPENDIX

## A.1 INFERENCE AND LEARNING ALGORITHMS FOR HMM-GLM

### A.1.1 FORWARD-BACKWARD INFERENCE

In this part, we compute the posterior probability given the old parameter $\theta^{\text{old}}$, which is the E-step of the EM algorithm. Define

$$\begin{cases} \gamma_{z_t}(t) := p(z_t|\boldsymbol{X};\theta^{\text{old}}) \\ \xi_{z_{t-1},z_t}(t) := p(z_{t-1},z_t|\boldsymbol{X};\theta^{\text{old}}) \end{cases}, \tag{9}$$

where $z_t$ indexes one of the $S$ different states.

Define

$$\begin{cases} \alpha_{z_t}(t) := p(\boldsymbol{x}_1,\ldots,\boldsymbol{x}_t,z_t) \\ \beta_{z_t}(t) := p(z_{t+1},\ldots,z_T|\boldsymbol{x}_1,\ldots,\boldsymbol{x}_t,z_t) \end{cases}, \tag{10}$$

and we have

$$\begin{aligned} \gamma_{z_t}(t) =& \frac{p(\boldsymbol{X},z_t)}{p(\boldsymbol{X})} \\ =& \frac{p(\boldsymbol{x}_1,\ldots,\boldsymbol{x}_t,z_t)p(\boldsymbol{x}_{t+1},\ldots,\boldsymbol{x}_T|\boldsymbol{x}_1,\ldots,\boldsymbol{x}_t,z_t)}{p(\boldsymbol{X})} \\ =& \frac{\alpha_{z_t}(t)\beta_{z_t}(t)}{p(\boldsymbol{X})}. \end{aligned} \tag{11}$$

$$\begin{aligned} \alpha_{z_t}(t) =& p(\boldsymbol{x}_1,\ldots,\boldsymbol{x}_t,z_t) \\ =& p(\boldsymbol{x}_t|\boldsymbol{x}_1,\ldots,\boldsymbol{x}_{t-1},z_t)p(\boldsymbol{x}_1,\ldots,\boldsymbol{x}_{t-1},z_t) \\ =& p(\boldsymbol{x}_t|\boldsymbol{x}_1,\ldots,\boldsymbol{x}_{t-1},z_t)\sum_{z_{t-1}} p(\boldsymbol{x}_1,\ldots,\boldsymbol{x}_{t-1},z_{t-1},z_t) \\ =& p(\boldsymbol{x}_t|\boldsymbol{x}_1,\ldots,\boldsymbol{x}_{t-1},z_t)\sum_{z_{t-1}} p(\boldsymbol{x}_1,\ldots,\boldsymbol{x}_{t-1},z_t|z_{t-1})p(z_{t-1}) \\ =& p(\boldsymbol{x}_t|\boldsymbol{x}_1,\ldots,\boldsymbol{x}_{t-1},z_t)\sum_{z_{t-1}} p(\boldsymbol{x}_1,\ldots,\boldsymbol{x}_{t-1}|z_t,z_{t-1})p(z_t|z_{t-1})p(z_{t-1}) \\ =& p(\boldsymbol{x}_t|\boldsymbol{x}_1,\ldots,\boldsymbol{x}_{t-1},z_t)\sum_{z_{t-1}} p(\boldsymbol{x}_1,\ldots,\boldsymbol{x}_{t-1}|z_{t-1})p(z_{t-1})p(z_t|z_{t-1}) \\ =& p(\boldsymbol{x}_t|\boldsymbol{x}_1,\ldots,\boldsymbol{x}_{t-1},z_t)\sum_{z_{t-1}} \alpha_{z_{t-1}}(t)p(z_t|z_{t-1}) \end{aligned} \tag{12}$$

with the initial condition

$$\alpha_{z_1}(1) = p(\boldsymbol{x}_1,z_1) = p(z_1)p(\boldsymbol{x}_1|z_1). \tag{13}$$

$$\begin{aligned} \beta_{z_t}(t) =& p(\boldsymbol{x}_{t+1},\ldots,\boldsymbol{x}_T|\boldsymbol{x}_1,\ldots,\boldsymbol{x}_t,z_t) \\ =& \sum_{z_{t+1}} p(\boldsymbol{x}_{t+1},\ldots,\boldsymbol{x}_T,z_{t+1}|\boldsymbol{x}_1,\ldots,\boldsymbol{x}_t,z_t) \\ =& \sum_{z_{t+1}} p(\boldsymbol{x}_{t+1},\ldots,\boldsymbol{x}_T|\boldsymbol{x}_1,\ldots,\boldsymbol{x}_t,z_{t+1},z_t)p(z_{t+1}|\boldsymbol{x}_1,\ldots,\boldsymbol{x}_t,z_t) \\ =& \sum_{z_{t+1}} p(\boldsymbol{x}_{t+1},\ldots,\boldsymbol{x}_T|\boldsymbol{x}_1,\ldots,\boldsymbol{x}_t,z_{t+1})p(z_{t+1}|z_t) \\ =& \sum_{z_{t+1}} p(\boldsymbol{x}_{t+2},\ldots,\boldsymbol{x}_T|\boldsymbol{x}_1,\ldots,\boldsymbol{x}_t,\boldsymbol{x}_{t+1},z_{t+1})p(\boldsymbol{x}_{t+1}|\boldsymbol{x}_1,\ldots,\boldsymbol{x}_t,z_{t+1})p(z_{t+1}|z_t) \\ =& \sum_{z_{t+1}} \beta_{z_{t+1}}(t+1)p(\boldsymbol{x}_{t+1}|\boldsymbol{x}_1,\ldots,\boldsymbol{x}_t,z_{t+1})p(z_{t+1}|z_t) \end{aligned} \tag{14}$$

with the initial condition $\beta_{z_T}(T) = 1$, since in Eq. 11

$$\gamma_{z_T}(T) = p(z_T|\boldsymbol{X}) = \frac{p(\boldsymbol{X}, z_T)\beta_{z+T}(T)}{p(\boldsymbol{X})} \equiv \frac{p(\boldsymbol{X}, z_T)}{p(\boldsymbol{X})}. \tag{15}$$

If we sum both sides of Eq. 11,

$$1 = \sum_{z_t} \gamma_{z_t}(t) = \frac{\sum_{z_t} \alpha_{z_t}(t)\beta_{z_t}(t)}{p(\boldsymbol{X})} \implies p(\boldsymbol{X}) = \sum_{z_t} \alpha_{z_t}(t)\beta_{z_t}(t), \tag{16}$$

and we can simply use $p(\boldsymbol{X}) = \sum_{z_T} \alpha_{z_T}(T)$ when $t = T$.

$$\begin{aligned}
\xi_{z_{t-1},z_t}(t) =& p(z_{t-1}, z_t|\boldsymbol{X}) \\
=& \frac{p(\boldsymbol{X}|z_{t-1}, z_t)p(z_{t-1}, z_t)}{p(\boldsymbol{X})} \\
=& \frac{p(\boldsymbol{X}|z_{t-1}, z_t)p(z_t|z_{t-1})p(z_{t-1})}{p(\boldsymbol{X})} \\
=& \frac{p(\boldsymbol{x}_t, \ldots, \boldsymbol{x}_T|\boldsymbol{x}_1, \ldots, \boldsymbol{x}_{t-1}, z_{t-1}, z_t)p(\boldsymbol{x}_1, \ldots, \boldsymbol{x}_{t-1}|z_{t-1}, z_t)p(z_t|z_{t-1})p(z_{t-1})}{p(\boldsymbol{X})} \\
=& \frac{p(\boldsymbol{x}_{t+1}, \ldots, \boldsymbol{x}_T|\boldsymbol{x}_1, \ldots, \boldsymbol{x}_t, z_{t-1}, z_t)p(\boldsymbol{x}_t|\boldsymbol{x}_1, \ldots, \boldsymbol{x}_{t-1}, z_{t-1}, z_t)\alpha_{z_{t-1}}(t-1)p(z_t|z_{t-1})}{p(\boldsymbol{X})} \\
=& \frac{p(\boldsymbol{x}_{t+1}, \ldots, \boldsymbol{x}_T|\boldsymbol{x}_1, \ldots, \boldsymbol{x}_t, z_t)p(\boldsymbol{x}_t|\boldsymbol{x}_1, \ldots, \boldsymbol{x}_{t-1}, z_t)\alpha_{z_{t-1}}(t-1)p(z_t|z_{t-1})}{p(\boldsymbol{X})} \\
=& \frac{\beta_{z_t}(t)p(\boldsymbol{x}_t|\boldsymbol{x}_1, \ldots, \boldsymbol{x}_{t-1}, z_t)\alpha_{z_{t-1}}(t-1)p(z_t|z_{t-1})}{p(\boldsymbol{X})}.
\end{aligned} \tag{17}$$

### A.1.2 BAUM–WELCH ALGORITHM

Now, we already have the posterior, and we proceed to the M-step of the EM algorithm.

$$p(\boldsymbol{X}, \boldsymbol{z}; \theta) = p(z_1; \theta)\left[\prod_{t=2}^{T} p(z_t|z_{t-1}; \theta)\right]\prod_{t=1}^{T} p(\boldsymbol{x}_t|\boldsymbol{x}_1, \ldots, \boldsymbol{x}_{t-1}, z_t; \theta). \tag{18}$$

$$\ln p(\boldsymbol{X}, \boldsymbol{z}; \theta) = \ln p(z_1; \theta) + \sum_{t=2}^{T} \ln p(z_t|z_{t-1}; \theta) + \sum_{t=1}^{T} \ln p(\boldsymbol{x}_t|\boldsymbol{x}_1, \ldots, \boldsymbol{x}_{t-1}, z_t; \theta). \tag{19}$$

Notice that

$$\begin{aligned}
Q(\theta, \theta^{\text{old}}) =& \sum_{z_1=1}^{S} \gamma_{z_1}(1) \ln p(z_1; \theta) + \sum_{t=2}^{T} \sum_{z_{t-1}=1}^{S} \sum_{z_t=1}^{S} \xi_{z_{t-1},z_t}(t) \ln p(z_t|z_{t-1}; \theta) \\
&+ \sum_{t=1}^{T} \sum_{z_t=1}^{S} \gamma_{z_t}(t) \ln p(\boldsymbol{x}_t|\boldsymbol{x}_1, \ldots, \boldsymbol{x}_{t-1}, z_t; \theta).
\end{aligned} \tag{20}$$

Problems regarding the scaling factor in the forward-backward algorithm for numerical stability and the Viterbi algorithm for predicting the most probable hidden sequence are identical to the plain HMM, which can be found in (Bishop & Nasrabadi, 2006).

## A.2 THRESHOLD

Fig. 6 shows the balanced accuracy of the adjacency matrices and the adjacency prior matrix by thresholding as a function of the threshold varying from 0 to 0.5. The plots demonstrate that, in general, the accuracy is very sensitive to the threshold.

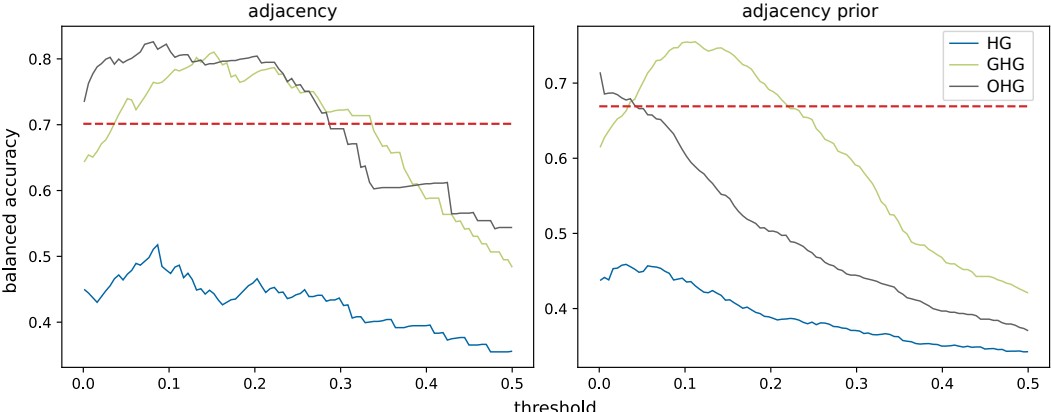

Figure 6: Using different thresholds to discretize the weight to obtain the adjacency and adjacency prior. The straight dashed red line is the balanced accuracy of $A_0$ obtained by OHG directly.

## A.3 SYNTHETIC DATASET

Fig. 7 shows the state inference of all models on one of the synthetic spike trains.

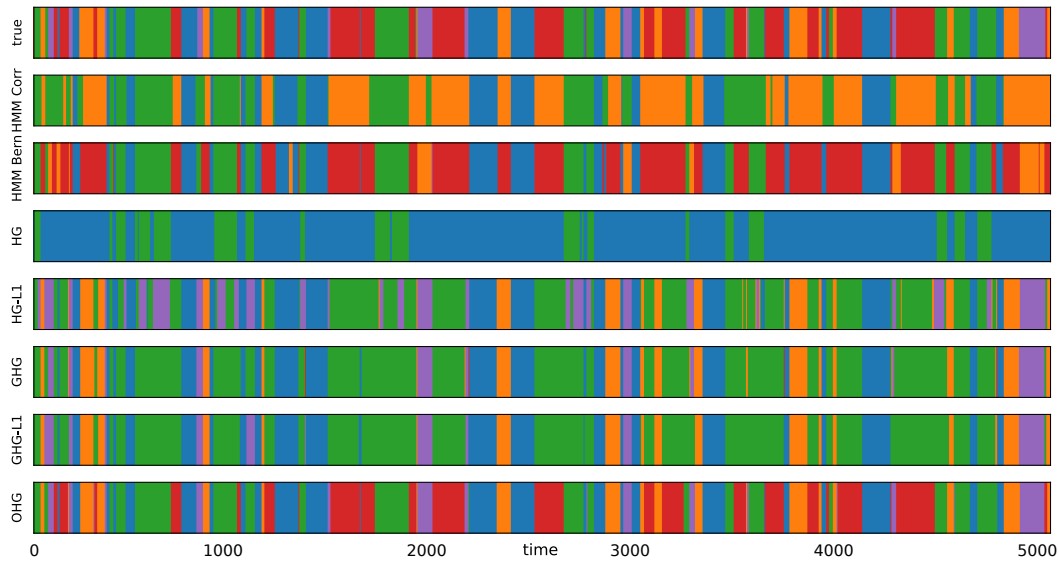

Figure 7: The inferred state sequences of all models applied to the one trial of the synthetic spike train data. Different colors represent different states.

## A.4 PFC-6 DATASET

### A.4.1 STATES VISUALIZATION OF CONSECUTIVE TRIALS

Fig. 8 shows the inferred state sequences of all models on trials 15-24.

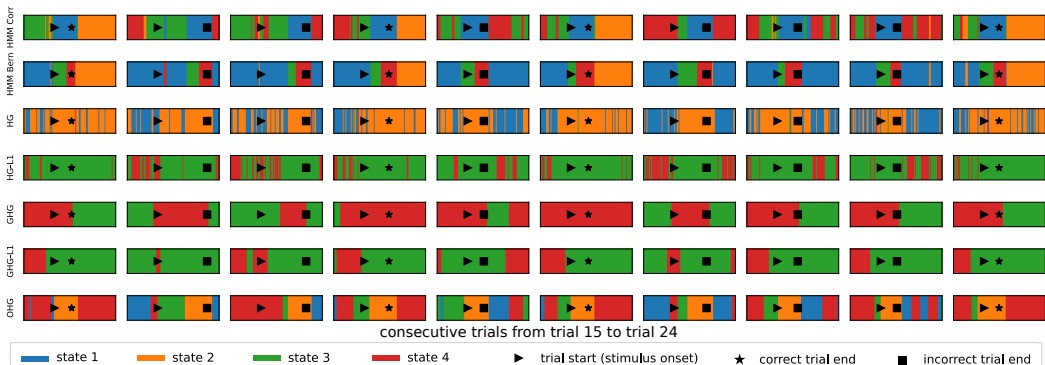

Figure 8: The inferred state sequences of all models applied to consecutive trials of the PFC-6 spike train data.

### A.4.2 DECODING TASK INFORMATION FROM INFERRED LATENT STATES

To further confirm our interpretation regarding the inferred states from OHG, we use logistic regression to decode the correctness of each trial from the inferred states. Specifically, each trial is viewed as a data point in logistic regression. The input variable is the one-hot representation of the inferred hidden states of size $S \times T$. We use the one-hot representation since the state in each time bin is a categorical variable. The output is a binary variable, representing the correctness of a trial. We use 2/3 trials to train the logistic regression and test on the remaining 1/3 trials.

Fig. 9 shows that the logistic regression fitted to the inferred states from OHG obtains the highest decoding accuracy. This means the inferred states from OHG do include enough information related to the correctness of each trial. From the interpretation in the main content, we mention that the animal will enter state 4 if it is a correct trial because of obtaining the reward at the correct target location. This is consistent with the positive coefficients of state 4 at the end period of trials, indicated by the black square in Fig. 9.

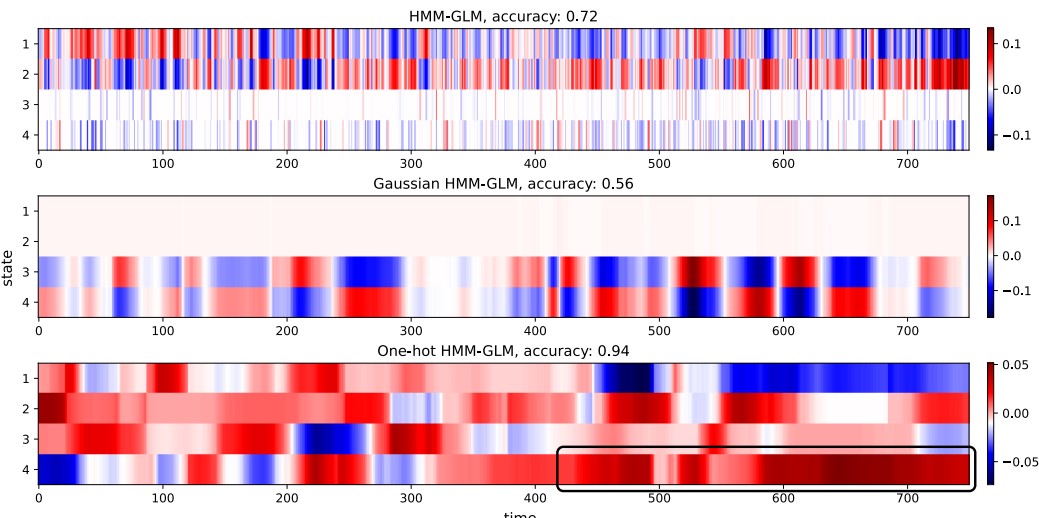

Figure 9: Coefficient matrices of logistic regression models correctness = LogisticRegression(inferred states) and the corresponding accuracy.

To further confirm the irreplaceable role of the inferred latent states in predicting trials' correctness, we train a multilayer perceptron (MLP) neural network with one hidden layer of size 100 (we tried different MLP configurations and selected the best one) to predict the correctness of each trial directly using the neural spike train as the input. The test accuracy is only 0.72, which is significantly worse than OHG. This implies that OHG plays a very important role in summarizing the task information from the neural spike train, similar to the irreplaceable role of the convolutional layers in CNN.

### A.4.3  OHG WITH UP TO 20 STATES

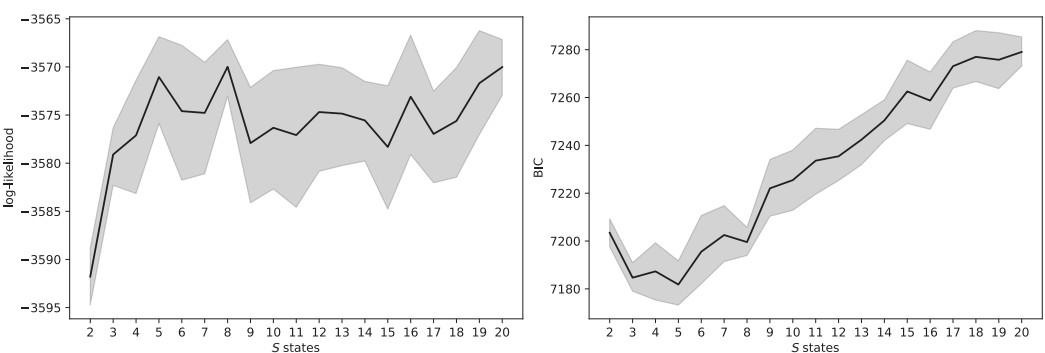

Figure 10: OHG's log-likelihood (left) and Bayesian information criterion (BIC) (right) w.r.t. up to 20 number of states. The performance does not increase significantly and becomes flat when $S \geqslant 5$.

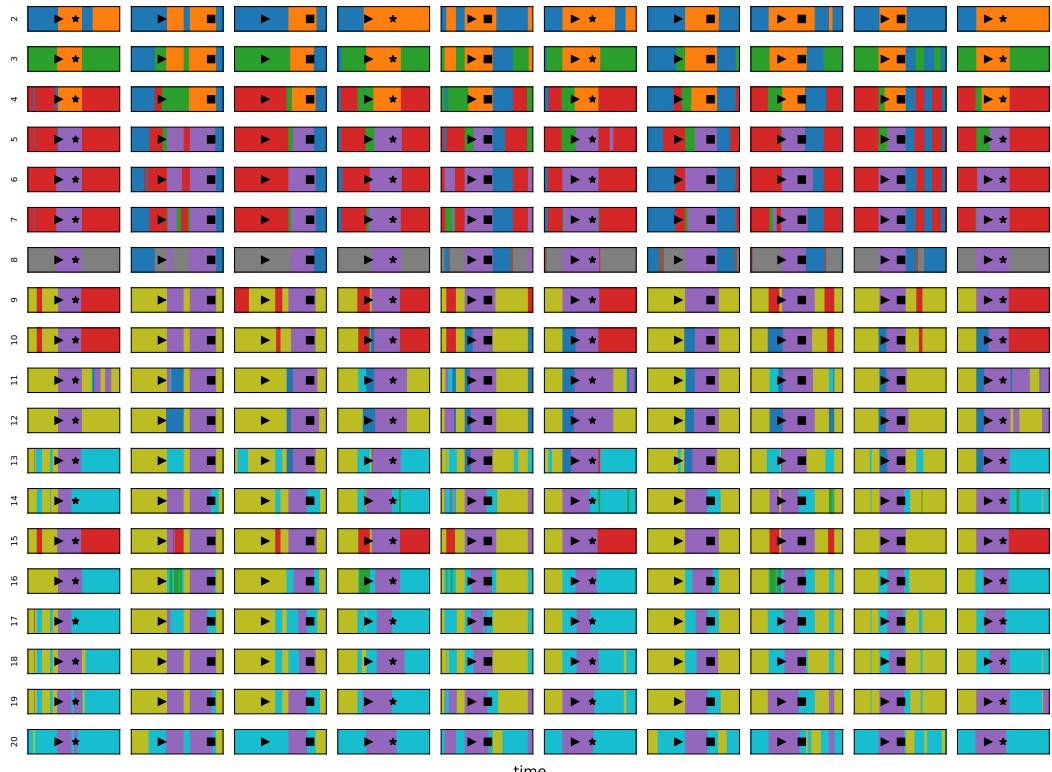

Figure 11: Inferred states of OHG with up to 20 number of states. Although more numbers of states are assumed, only around four effective states are learned and the interpretations are all similar to that of the 4 states explained in the main content.

## A.5 BARREL CORTEX DATA WITH 5 HIDDEN STATES

As noted in the main text, OHG exhibits increasing test log-likelihood with an increasing number of states $S$. When $S = 5$, there were typically 2 or 3 dominant states inferred from OHG, with the other states being inferred only rarely across the sessions. Fig. 12 shows an example of a trial with $S = 5$. OHG exhibits one dominant hidden state (blue) with the other states being inferred for short intervals of duration 0.1-0.3 s, showing complex activation patterns in the vicinity of whisker contacts. The corresponding weight and adjacency matrices are shown in Fig. 13. Further analysis is needed to determine the significance of such states.

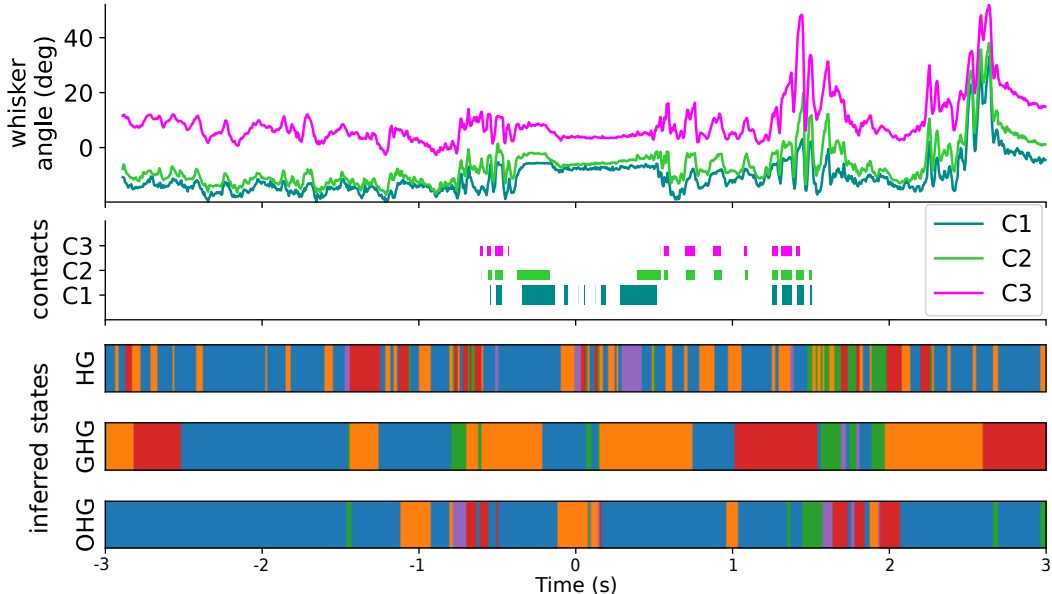

Figure 12: Example trial from barrel cortex data with $S = 5$ hidden states.

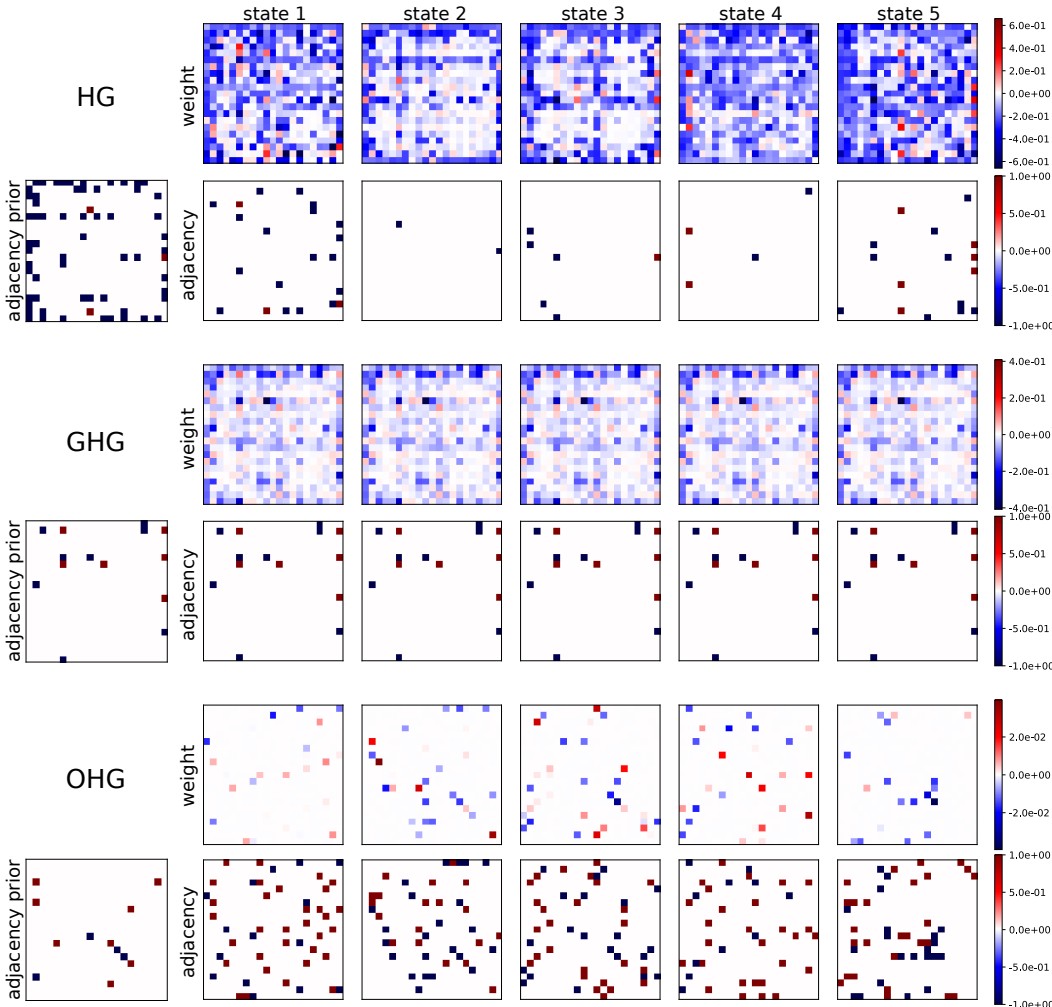

Figure 13: Weights, adjacencies, and adjacency prior for each state of HG, GHG, and OHG applied to the barrel cortex data session shown in Fig. 12.

### A.6 SUBPOPULATION (MULTI-REGION) PROBLEMS

Although all HMM-GLMs in this paper including our newly proposed GHG and OHG only work for one global prior, it is still possible to apply the model to multi-region data. For example, $C$ groups of neurons $N_1, N_2, \ldots, N_C$ evolve largely independently and only rarely communicate. Further, assume each group of neurons has its own global structure and its own number of states $S_1, S_2, \ldots, S_C$. Then, we can reduce such a problem into a model with one single globally shared structure $\mathrm{diag}(A_{1,0}, A_{2,0}, \ldots, A_{C,0})$. Then, there will be $S = S_1 \times S_2 \times \cdots \times S_C$ states for the whole groups of neurons in total.

For better illustration, we run a simple example on $C = 3$ subpopulations (groups) and each group has 5 neurons and 2 states. Therefore, we need a model of 15 neurons and 8 states to accommodate this configuration.

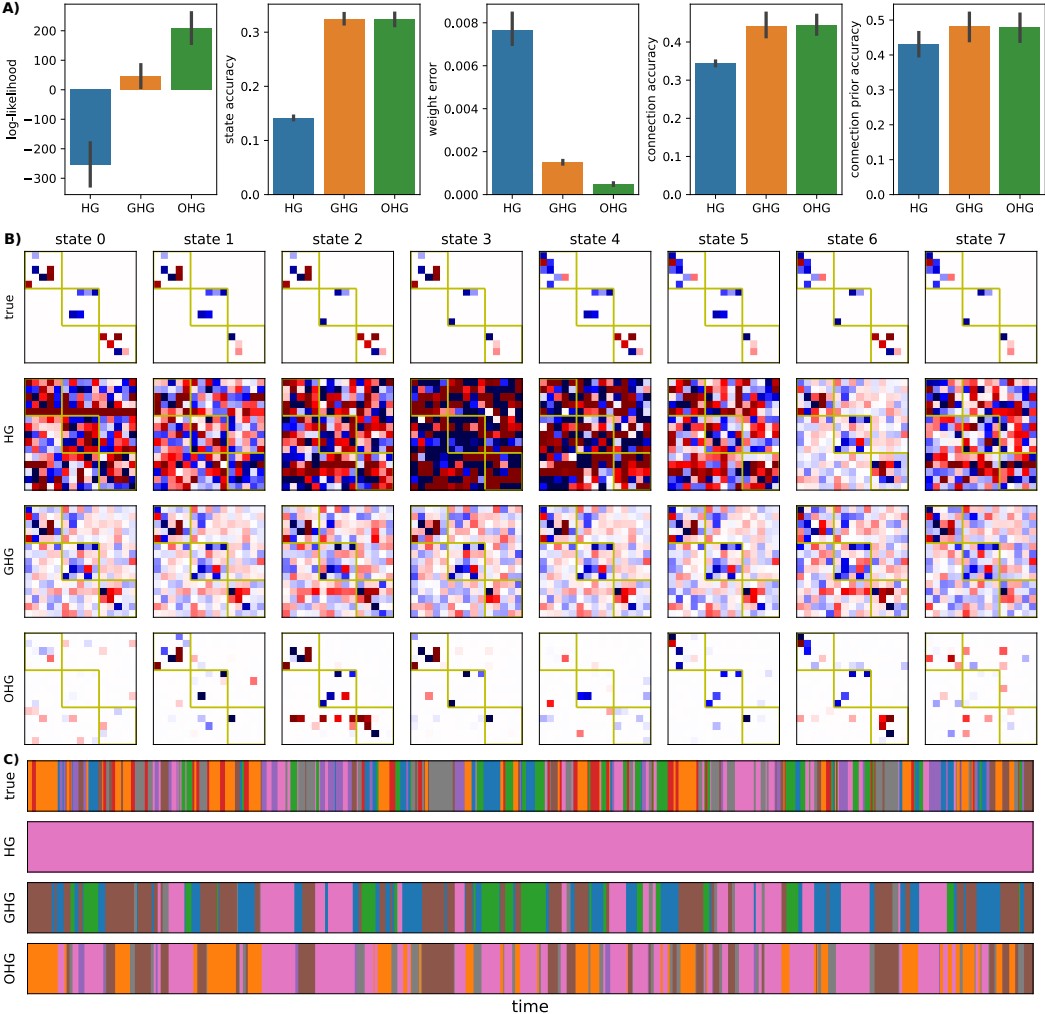

Figure 14: A) The quantitative results with 5 metrics for HG, GHG, and OHG. B) The learned weight matrices in all 8 states. The top-left block represents the weight matrices corresponding to group 1; the middle block represents the weight matrices corresponding to group 2; the bottom-right block represents the weight matrices corresponding to group 3. C) The true and inferred states by HG, GHG, and OHG. 8 colors represent 8 states. Particularly, at each time bin $t$, if the state is $(s_1, s_2, s_3)$, then the overall state is the $s = 2^2 \times s_1 + 2^1 \times s_2 + 2^0 \times s_3$. I.e., $(s_1, s_2, s_3)$ is the binary encoding of the overall state $s$.

Fig. 14A shows that OHG is better than HG and GHG in terms of LL and weight error. For state accuracy, adjacency accuracy, and adjacency prior accuracy both GHG and OHG are better than

HG. However, since these models are not targeted tasks including subpopulations, the overall state accuracies for all models are worse than usual. The reason is that the state complexity of the reduced problem increases exponentially w.r.t. the number of groups. Fig. 14B shows the true weight matrices in all 8 combination states (using binary encoding) and the learned weight matrices from HG, GHG, and OHG. Consistent with the weight error reported in Fig. 14A, OHG is better than GHG and is better than HG. Particularly, OHG learns meaningful weight matrices in each group. For example, the top-left block of OHG in states 1, 2, and 3 corresponds to the state 0 weight matrix of group $c = 1$. The top-left block of OHG in states 5 and 6 corresponds to the state 1 weight matrix of group $c = 1$. Fig. 14C qualitatively shows that the inferred states from GHG and OHG are at an acceptable level compared with the true overall state transitions.

## A.7 Relationships to SLDS, rSLDS, and mp-rSLDS

Instead of considering state switching on neural connectivities as our HMM-GLM-based models, SLDS (Ackerson & Fu, 1970; Chang & Athans, 1978; Hamilton, 1990; Ghahramani & Hinton, 1996; 2000; Murphy, 1998; Fox et al., 2008), rSLDS (Linderman et al., 2016; 2017), and mp-rSLDS (Glaser et al., 2020) consider state switching on underlying linear dynamics governing the observed spike trains. Besides, mp-rSLDS also considers incorporating prior information (such as anatomy) into the linear mappings in different states. However, its shared prior is a known hyperparameter, but the shared global prior in our GHG and OHG are learnable.

Although SLDS, rSLDS, and mp-rSLDS don't learn neural connectivities in different states explicitly, both SLDS-based models and HMM-GLM-based models learn discrete state switches from spike trains. Since we don't consider multi-region or subpopulations in this work, we compare the inferred states from OHG with those from SLDS and rSLDS on the PFC-6 dataset.

Fig. 15 shows the inferred states from SLDS and rSLDS exhibit fast-switching phenomena, which hinders their interpretability. This result might imply that state switching over neural connectivities could be an important assumption that should be taken into account when dealing with spike train data collected from multi-stage experiments.

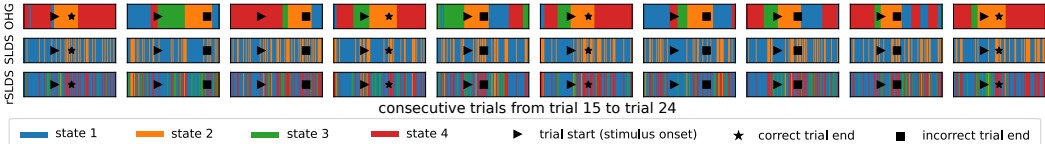

Figure 15: The inferred states from OHG, SLDS, and rSLDS applied to consecutive trials of the PFC-6 spike train data.

## A.8 SCALABILITY TO LARGE NUMBERS OF NEURONS

From the synthetic dataset, we observe that the state accuracies of all HMM-GLM-based models start to drop when there are 40-50 neurons. To this scale, there will be more than $S \times [1600, 2500]$ edges (connections) that need to be learned in the network, which introduces challenges in determining the correct state switches. Under such situations, the effectiveness of both the E-step and the M-step in the algorithm would be mutually influenced by each other. However, this does not mean that the HMM-GLM-based models cannot be applied to real-world datasets. It might still be worth trying HMM-GLM-based models even if the data consists of a large number of neurons, since Fig. 16 shows that the per-neuron log-likelihood does not drop when increasing the number of neurons. This means that HMM-GLM-based models (especially OHG) can still predict firing rates effectively for spike train data reconstruction. We would like to view this as an important future direction.

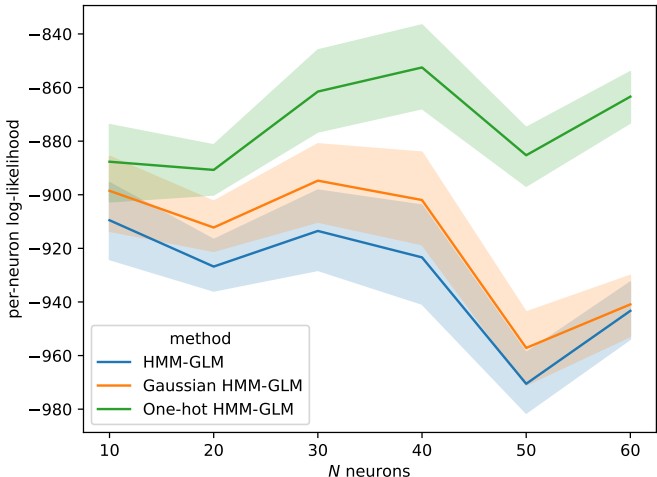

Figure 16: The per-neuron log-likelihoods of HG, GHG, and OHG w.r.t the number of neurons.

## A.9 PRIOR ON THE TRANSITION MATRIX

An intuitive way of suppressing the fast state switches in HG is to add an L2 regularization term on the transition matrix. However, Fig. 17 shows that when the number of state switches is suppressed, the inferred states are still meaningless. Specifically, most switches happen within a short duration, which looks like nothing but noisy state switches. Only one major state governs the whole trial.

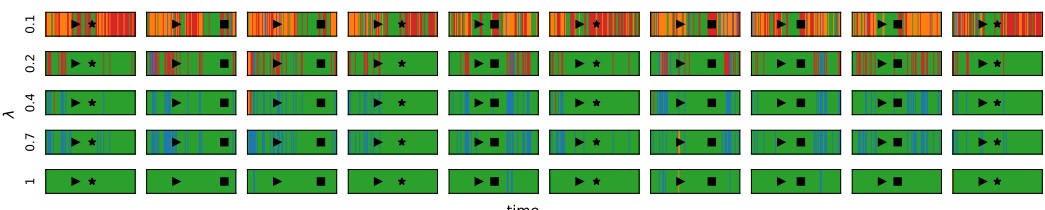

Figure 17: The inferred states from HGs with different regularization terms applied to consecutive trials of the PFC-6 spike train data.

