# OpenReview forum: "One-hot Generalized Linear Model for Switching Brain State Discovery"
_ICLR.cc/2024/Conference — ICLR 2024 poster_

### Official Review · Reviewer_G7kx · 2023-10-31

**Soundness:** 4 excellent
**Presentation:** 4 excellent
**Contribution:** 4 excellent
**Rating:** 8
**Confidence:** 5

**Summary:**

The authors address the challenge of modeling dynamic functional neural interactions. They note that existing methods often lack biological plausibility, primarily because they don't account for the influence of anatomical structures on functional neural interactions. To rectify this, the authors introduce a one-hot prior to the GLM model. The method was evaluated on one synthesized dataset and two real-world datasets, achieving state-of-the-art results.

**Strengths:**

This paper is technically robust. The underlying problem is well-defined and builds upon a lineage of substantial research. Drawing insights from neuroscience, the authors convincingly argue that anatomical structures influence dynamic functional neural interactions. Their approach to address this hypothesis is adeptly framed, straightforward, and effective. The evaluation is comprehensive, encompassing a broad spectrum of models related to the problem, and it's tested across varied datasets. The inclusion of the whisking dataset is particularly intriguing, and the visual illustrations enhance clarity. Overall, this paper is commendable and would be a valuable contribution to the ICML community, showcasing the intersections of machine learning and neuroscience research.

**Weaknesses:**

(1) While the overall presentation of the paper is commendable, there is room for improvement in Sections 2 and 3. These sections could benefit from more intuitive and lucid explanations accompanying the mathematical equations, making it more accessible for readers.

(2) I believe the prior work by Glaser et al. [1] deserves acknowledgment. It might also be valuable to include it in the comparative models, given that their focus on cluster (population) structures aligns with the theme of underlying structures.

[1] Glaser, Joshua, et al. "Recurrent switching dynamical systems models for multiple interacting neural populations." Advances in Neural Information Processing Systems 33 (2020): 14867-14878.

**Questions:**

I'm keen to understand the authors' future direction and insights drawn from this research. Does incorporating an increasing number of biological constraints into models always lead to better outcomes? Or are there potential trade-offs to be mindful of? Going forward, are the authors considering other factors that might influence interactions? For instance, within an E-I balanced network, given identical anatomical structures and brain states, interactions could vary based on the stage and phase of short-term synaptic depression. This suggests that intrinsic governing features could arise when adding more biological constraints or features. I'd appreciate the authors' perspective on this.

---

> ### Author Response · Authors · 2023-11-16
> **Reply to G7kx**
>
> Dear Reviewer G7kx,
>
> Thank you very much for your time and valuable comments on our paper. We appreciate your recognition of the strengths of our paper. Hopefully, the following responses could resolve most of your concerns and answer your questions.
>
> ### Weaknesses
> 1. Following the reviewer's suggestion, we have reorganized the Methods section to improve readability for readers. We have changed the order of the presentation to introduce the models in order of increasing complexity. Furthermore, we have added a subsection detailing the relation between GHG and OHG. We hope these changes have made the presentation more intuitive for readers. Please check the current latest revision.
> 2. We agree that the work by Glaser et al. (2020) should be mentioned in the paper.
>     * The citations regarding SLDS, rSLDS, and mp-SLDS are included in the introduction section.
>     * Relationships to SLDS-based methods: Instead of considering state switching on neural connectivities as our HMM-GLM-based models, SLDS, rSLDS, and mp-rSLDS consider state switching on underlying linear dynamics governing the observed spike trains. Besides, mp-rSLDS also considers incorporating prior information (such as anatomy) into the linear mappings in different states. However, their shared priors are known hyperparameters, but the shared global priors in our GHG and OHG are learnable.
>     * We have finished the comparison of our OHG with SLDS-based methods shown in Fig. 16 and Appendix A.7 in our latest revision. Specifically, the inferred states from SLDS-based methods exhibit fast-switching phenomena, which hinders their interpretability. This result might imply that state switching over neural connectivities could be an important assumption of an effective component that should be taken into account when dealing with spike train data collected from multi-stage experiments.
>
>
> ### Questions
> **Future Directions**: We greatly appreciate the reviewer's perspective shown through their questions. Indeed, placing a biologically motivated constraint on the structural connectivity with the OGH improved inference of both state and functional connectivity; our results thus indicate that including relevant biological constraints can lead to a better outcome. On the other hand, adding too many biological details can lead to overparameterization of the model, so each added degree of complexity should be carefully chosen and evaluated. We agree that one promising extension would be to add effects of short-term synaptic potentiation and depression, which is biologically well-characterized but not included in our models. Adding constraints to control the E-I balance could also be beneficial. We leave these promising topics to a future study. Another extension would be to add different assumptions about the network structure, for example, the inclusion of multiple regions with dense intra-regional connectivity and sparse inter-regional connectivity. This direction was suggested by Reviewer hZX1 and is also explored in the aforementioned study by Glaser et al. Please check Fig. 9 in Appendix 4 in the current revision for the result regarding subpopulations of neurons.
>
> We hope that this response addresses the majority of your concerns and questions. Your feedback is valuable, so please don't hesitate to provide additional comments or ask further questions. Thank you again for your time and valuable feedback!

---

> > ### Comment · Reviewer_G7kx · 2023-11-21
> >
> > Thank you to the authors for addressing my questions and concerns, and for incorporating the reference I mentioned for comparison. I believe it is an interesting and good paper to the ICLR community.

---

### Official Review · Reviewer_zShZ · 2023-10-31

**Soundness:** 2 fair
**Presentation:** 3 good
**Contribution:** 2 fair
**Rating:** 3
**Confidence:** 4

**Summary:**

This paper proposes a prior-informed state-switching generalized linear model with hidden Markov models (HMM-GLM) called one-hot HMM-GLM (OHG), capable of estimating dynamically changing functional interactions under different states. Learnable priors are introduced to capture the state-constant interaction and reveal the underlying anatomical connectome. Experiments on simulated data demonstrated its effectiveness and practical applications achieved interpretable interaction structures and hidden states with the highest predictive likelihood.

**Strengths:**

1. This paper proposed a novel OHG framework to estimate time-varying functional interaction in multi-state neural systems. The one-hot prior yielded better connectivity patterns and hidden states and provided more biological plausibility.
2. This paper provided detailed algorithms of the proposed model and conducted extensive experiments on both synthetic and real neural datasets to demonstrate its superiority.

**Weaknesses:**

1. This paper seems to propose two frameworks: the naïve one is GHG and the effective one is OHG. What’s the relationship between them? In the abstract, the authors only mention the two priors (Gaussian and one-hot) without the names of the frameworks. In the conclusion, only OHG is mentioned. Thus, it is confusing.
2. In the method, the authors first describe OHG and then introduce GHG. They are both variants of HMM-GLM but OHG outperforms GHG. Thus, the order seems unreasonable. What’s more, the experimental results showed that GHG was unable to achieve this paper’s goal. Then what’s the value of GHG?

**Questions:**

1. As shown in Table 2, the results of different numbers of states were similar to that of one-state GLM. It can be explained that global static connection patterns dominate functional interactions in all states as mentioned in the manuscript. Then was the state division biologically reasonable? Perhaps only the features of the global prior were extracted or there was only one state.
2. The experiments fixed the generative hyperparameters and claimed that this set was noninformative priors and insensitive to different datasets. Is there any support for this declaration?

---

> ### Author Response · Authors · 2023-11-16
> **Reply to zShZ (1/2)**
>
> Dear Reviewer zShZ,
>
> Thank you very much for your time and valuable comments on our paper. We highly appreciate your recognition of the strengths of our paper. Hopefully, the following responses could resolve most of your concerns and answer your questions.
>
> ### Weaknesses
> 1. **Mentioning GHG and OHG, and the value of GHG**: Thanks for your valuable suggestion. We have modified our abstract, method, result, and conclusion parts to include more discussion on GHG. Specifically:
>     * **Abstract**: We include the names of the two frameworks (two variants of HMM-GLM): GHG and OHG explicitly.
>     * **Method**: We add an extra comparison between OHG and GHG. Compared with OHG, **GHG serves as an intermediate model with a straightforward prior directly on the weights** representing shared global connectivity but without the one-hot decomposition.
>     * **Experiment results**: In the experimental evaluation section, GHG is not bad, but still worse than OHG, validating that the one-hot decomposition component in OHG does account for the better performance achieved by OHG.
>     * **Conclusion**: We add new sentences mentioning the role of GHG. I.e., GHG serves as an intermediate model with shared prior directly on weight matrices without (one-hot) strength-connection decomposition, confirming that such a decomposition is critical for the success of multi-state inference.
> 2. **Order of introducing GHG and OHG**: We thank the reviewer for pointing out this issue. The value of GHG is not in that it outperforms the OHG, but lies in its role as an intermediate model. The GHG is the simplest way to add a prior to the state-dependent weight matrices $\boldsymbol W_s$. The OHG has an additional layer of complexity in that it decomposes the weight matrices into a product of a discrete connection matrix and a strength matrix, allowing the model to specifically constrain the structural connectivity. The value of including the GHG in our study is thus not that the GHG outperforms the OHG, but rather that it allows us to examine whether the structure-strength decomposition of the OHG is necessary, or whether a simple Gaussian prior over the weight matrix is enough. The fact that the OHG significantly outperforms the GHG in both state prediction and inference of the underlying functional connectivity shows that the structural decomposition method provides additional benefits that a simple prior cannot.   &nbsp;&nbsp;&nbsp;&nbsp;     However, we understand that the presentation of the Methods may be confusing in that the final method introduced is an intermediate model. Accommodating the reviewer's comment, we rearranged the Methods to introduce the models in increasing order of complexity: The GHG is now introduced before the OHG, which is introduced last. Furthermore, in order to clarify the role of GHG in our study we added a subsection in the Methods titled "The relationship between GHG and OHG".

---

> > ### Author Response · Authors · 2023-11-16
> > **Reply to zShZ (2/2)**
> >
> > ### Questions
> >
> > 1. **One-state GLM vs OHG**: As the reviewer points out, the absolute difference between the one-state GLM and the OHG is not large. Nevertheless, we believe that the results of the multi-state OHG are significant for the following reasons. First, the difference in the log-likelihood of the OHG model compared to the one-state GLM is statistically clearly significant, as the standard error in the log-likelihood for the 5-state OHG is ~0.03 while the OHG has an increased log-likelihood of ~0.64. Furthermore, the latent states predicted by the multi-state OHG have a clear interpretation: as shown in Fig. 4, the states correspond to different behavioral stages as the animal performs the task. The multi-state OHG is thus capable of detecting intra-trial changes in functional connectivity that meaningfully reflect behavior. The fact that other multi-state models perform worse than the one-state GLM indicates that the way in which the prior is imposed is critical to obtaining reliable state prediction, i.e., a simple Gaussian prior is not enough and the one-hot component is necessary. So the multiple states are necessary, but only provide useful information with structural prior as in OHG.
> >
> > 2. **Noninformative priors**:
> >     * We agree with the reviewer that a more principled way of choosing hyperparameters of the OHG via cross-validation would be desirable. In the current treatment, we manually chose a flat prior (i.e., large variance/heavy tail) which reflects our prior knowledge of the parameters. We did this because there was no principled manner in which to select a prior distribution for weights, biases, and other model parameters. Following the reasoning laid by Zeller [1], we selected a noninformative, flat prior which expresses our lack of prior information about the parameters. This approach is akin to selecting a small regularization coefficient $\lambda$ in a ridge or lasso regression.
> >     * To provide further justification for this hyperparameter choice, we performed additional experiments comparing the performance of OHG under different prior variances of  $\tilde w_{s,n\gets n'}$. The larger the $1/\sigma_w$, the stronger the prior. The following shows that the average log-likelihood on 10 different datasets decreases as the prior becomes stronger. This implies that it would be better to set a conservative weak prior to allow the model to learn parameters more reliant on data, rather than a strong prior assumption.
> >
> >     | $\propto \frac{1}{\sigma_w}$ | 0 (absolute noninformative prior) | 0.001 | 0.01 | 0.1 | 1 | 10 | 100 |
> >     | - | - | - | - | - | - | - | - |
> >     | average log-likelihood | 2.26 | 2.03 | 1.81 | 1.37 | 1.34 | 0.97 | 0.17 |
> >    * We believe that this result provides some evidence that our choice of hyperparameter is reasonable. However, as noted above, a more rigorous treatment with cross-validation would be desirable. We thank the reviewer for the suggestion and in the interest of preserving the scope of the paper, leave this for a future study.
> >
> > [1] Zellner, Arnold (1971). "Prior Distributions to Represent 'Knowing Little'". An Introduction to Bayesian Inference in Econometrics. New York: John Wiley & Sons. pp. 41–53.
> >
> > We hope that this response addresses the majority of your concerns and questions. The clarity of our paper is improved and our ideas are conveyed better now. Your feedback is valuable, so please don't hesitate to provide additional comments or ask further questions. Besides, we have made the necessary modifications as per your valuable suggestions. Hopefully, our answers and revised paper could help you reevaluate our paper. Thank you again for your time and valuable feedback!

---

> > > ### Author Response · Authors · 2023-11-22
> > > **Sincerely looking forward to your feedback before deadline**
> > >
> > > Dear Reviewer zShZ
> > >
> > > Thank you very much again for your initial review of our paper. Given it is very close to the discussion deadline, and we are not sure whether you are satisfied with our answers and revised paper or whether there are still some concerns that have not been fully resolved, we would like to kindly remind you that we have answered all your questions and concerns mentioned in the weaknesses. Our latest revision provides a clearer picture of our work.  We are more than willing to hear your new feedback on them. We appreciate your time and invaluable feedback.
> > >
> > > Best,

---

> > ### Comment · Reviewer_G7kx · 2023-12-04
> >
> > Yes, putting both two methods is a good way to examine the necessity for the structure-strength decomposition since a simpler prior might be the first one come to everyone's mind. I am also looking forward to reviewer zShZ's response on that.

---

### Official Review · Reviewer_hZX1 · 2023-10-31

**Soundness:** 4 excellent
**Presentation:** 4 excellent
**Contribution:** 3 good
**Rating:** 8
**Confidence:** 4

**Summary:**

The paper discusses an extension of generalized linear models for a population of neurons used (binned spike trains under Poisson firing rate assumption)  that involves latent states and factorized latent-state-dependent inter-neuron connection weights.  The latter goes beyond previous work with a specific factorization that involves a mixture factor over at the simplex, which at its extremes provides a one-hot encoding that determines the existence of a connection and its sign (excitatory or inhibitory), and the state-dependent weight magnitude. Estimation of the parameters of this model requires an expectation maximization algorithm, which is briefly described. Baseline models from the literature and additional novel baselines are constructed by involving various aspects of the proposed approach. Results are presented for a synthetic experiment and two real-world data sets, with known task/stimulus/environmental timing.

**Strengths:**

The paper is an original contribution for GLM models of neuron spike trains. The method and results are well-presented and clear. The figures and equations are clear. A number of baselines are compared and the results are consistent.  From the results it would seem that the latent state inference is meaningful, this could be significant for neuroscientists who wish to study.

**Weaknesses:**

The synthetic study seems quite limited to the type of data the model is designed for (a single global state).  It is not clear to me how well it will work if the neurons are organized into groups with their own state dynamics (which evolve largely independently) and only rarely communicate. I.e. the topology of the network could be loose connections between tightly interconnected subnetworks.

A principled approach for the selection of the number of states is not discussed. At one point the paper mentions that the log-likelihood is higher with additional states although these states are rare: "there are many sessions with rarely occupied states, and the distinction
between states becomes subtle". This seems to be a flaw in the modeling if someone does not know how many true states. Should the reader be suggested to look at the distribution of states to decide? Perhaps a model selection criterion is needed.

Along similar lines, an analysis of the decoding of task information from the latent state would help understand in the real-world tasks the utility of the state estimate.

Questions of scaling could provide better significance:

How scalable is the model and/or the algorithm? New recording technology including optical calcium imaging can record from hundreds to close to thousands of neurons.  The number of neurons in the synthetic study could be ramped up to see this.

It is not clear how quickly can inference be performed after model fitting. If a neuroscientist wants to use the inferred state to control a stimulus is it possible to operate in real-time with a minimal delay?

**Questions:**

How would the number of states be selected in practice?

How scalable is the model in terms of subpopulations with their own dynamics?

How scalable is the model and algorithm in terms of the number of neurons?

How quickly can inference be done at run time?

---

> ### Author Response · Authors · 2023-11-17
> **Reply to hZX1 (1/2)**
>
> Dear Reviewer hZX1,
>
> Thank you very much for your time and valuable comments on our paper. We highly appreciate your recognition of the strengths of our paper. Hopefully, the following responses could resolve most of your concerns and answer your questions.
>
> * **Experiments on subpopulation configurations**: Although all HMM-GLMs in this paper including our newly proposed GHG and OHG only work for one global prior, it is still possible to apply the model to multi-region data. For example, $C$ groups of neurons $N_1,N_2,\dots,N_C$ evolve largely independently and only rarely communicate. Further, assume each group of neurons has its own global structure and its own number of states $S_1,S_2,\dots,S_C$. Then, we can reduce such a problem into a model with one single globally shared structure $\operatorname{diag}(\boldsymbol A_{1,0},\boldsymbol A_{2,0},\dots,\boldsymbol A_{C,0})$. Then, there will be $S = S_1\times S_2\times\dots\times S_C$ states for the whole groups of neurons in total. For better illustration, we run a simple example on $C=3$ subpopulations (groups) and each group has 5 neurons and 2 states. Therefore, we need a model of 15 neurons and 8 states to accommodate this configuration. The results are shown in Fig. 14 in Appendix 6 in the current revision. Although this is a feasible solution to using our HMM-GLMs framework to tackle subpopulations problem, we think it is better and more reasonable to directly develop a new model for subpopulations problem, and this could be one of our great future directions.
> * **Number of states**:
>     * In real-world scenarios, we don't know the true number of true states or there is even no definitive concept of absolute true states. Cross-validation is certainly a rigorous way to determine the number of states. However, increasing the number of states may increase the validation log-likelihood, as the OHG result in Tab. 2. Too many states may hurt interpretability. Therefore, there should be a trade-off between the validation log-likelihood and the interpretability. It might be better to select a turning point where the likelihood stops its rapid growth, and in the meantime, the number of states is not too large to be interpreted.
>     * For example, for $S>5$ in the PFC-6 dataset, the performance of OHG does not increase significantly and becomes flat (Fig. 10} in Appendix 4.3. Although more numbers of states are assumed, only around four effective states are learned and the interpretations are all similar (Fig. 11 in Appendix 4.3) to that of the 4 states explained in the main content.
>     * Generally speaking, the problem of determining a suitable number of states exists in most of the multi-state models, and this could be one of our important future directions.
> * **Decoding task information from inferred latent**:
>     * Thanks and we think this is a good suggestion. To further confirm our interpretation regarding the inferred states from OHG, we use a logistic regression model to decode the correctness of each trial from the inferred states. Specifically, each trial is viewed as a data point in logistic regression. The input variable is the one-hot representation of the inferred hidden states of size number of states $\times$ number of time bins. We use the one-hot representation since the state in each time bin is a categorical variable. The output is a binary variable, representing the correctness of a trial. We use 2/3 trials to train a logistic regression model and test on the remaining 1/3 trials. Fig. 9 in Appendix 4 shows that logistic regression fitted to the inferred states from OHG obtains the highest decoding accuracy. This means the inferred states from OHG do include enough information related to the correctness of each trial. From the interpretation in the main content, we mention that the animal will enter state 4 if it is a correct trial because of obtaining the reward at the correct target location. This is consistent with the positive coefficients of state 4 at the end period of trials, indicated by the black square in Fig. 9.
>     * To further confirm the irreplaceable role of the inferred latent in predicting trials' correctness, we train a multilayer perceptron (MLP) neural network with one hidden layer of size 100 (we tried different MLP configurations and selected the best one) to predict the correctness of each trial directly using the neural spike train as the input. The test accuracy is only 0.72, which is significantly worse than OHG. This implies that OHG plays a very important role in summarizing the task information from the neural spike train, similar to the irreplaceable role of the convolutional layer in CNN.
>     * Please check the details and results in Appendix 4 in the current revision.

---

> ### Author Response · Authors · 2023-11-17
> **Reply to hZX1 (2/2)**
>
> * **Scalable to the number of neurons**:
>     * The time complexity of the learning in one epoch is $\mathcal O(N^2TS)$ for all HMM-GLM-based methods (including HG, GHG, and OHG), where $N$ is the number of neurons, T is the number of time bins, and $S$ is the number of states. Therefore, the time increases quadratically to the number of neurons $N$ since there will be $\frac{N(N-1)}{2}$ edges in each state. However, since the parameter set of OHG is larger than GHG and larger than naive HG, the constant coefficient of OHG before the complexity term $N^2TS$ is larger than GHG and larger than naive HG. To understand the scalability of our algorithm to large numbers of neurons, we are running experiments on this and will keep you posted when the result can be accessed in the Appendix in our next revision.
>     * From the synthetic dataset, we observe that the state accuracies of all HMM-GLM-based methods start to drop when there are 40-50 neurons. To this scale, there will be more than $S\times [1600,2500]$ edges (connectivities) that need to be learned in the network, which introduces challenges to determining the correct state switches. Under such situations, the effectiveness of both the E-step and the M-step in the algorithm would be mutually influenced by each other. However, this does not mean that the HMM-GLM-based methods cannot be applied to real-world datasets. It might still be worth trying HMM-GLM-based methods even if the data consists of a large number of neurons, since Fig. 17 in Appendix A.8 shows that the per-neuron log-likelihood does not drop when increasing the number of neurons. This means the HMM-GLM-based models (especially OHG) is still to predict firing rates effectively for spike train data reconstruction. We would like to view this as an important future direction.
> * **Inference time**: Once the model parameter is learned, the inference should be very quick. Specifically, the time complexity to run the forward-backward algorithm to infer the state sequence is $\mathcal O(2ST)$. On the synthetic dataset with 5 states and 20 neurons, the running time of the forward-backward algorithm on a sequence of length 5000 time bins is about 1 second (on one core of the Intel Xeon Gold 6226 "Cascade Lake" @ 2.7Ghz CPU). Once the model parameters are learned from previous spike trains, we can rapidly infer the state sequence for the current spike train, and then use it to control the stimulus for the next trial.
>
> We hope that this response addresses the majority of your concerns and questions. Your feedback is valuable, so please don't hesitate to provide additional comments or ask further questions. Thank you again for your time and valuable feedback!

---

> > ### Comment · Reviewer_hZX1 · 2023-11-20
> >
> > I want to thank the authors for the additional work, which further impresses the contribution of the work.
> >
> > I have three questions with remaining time for discussion.
> >
> > 1. Is the relative performance (using the whole battery) on the multiple population case comparable to the synthetic example in the main body? It seems (perhaps because of the block structure) that there are more spurious correlations.
> >
> > 2. Wouldn't Bayesian information criterion provide a principled trade-off between increasing number of states and the validation likelihood? I agree the knee is clear in the plot, but a justified criterion for automatic model selection is helpful in practice.
> >
> > 3. It seems like the comparisons/baselines of HMM-GLM and SLDS have very fast latent state dynamics. Cannot this be remedied by a prior to bias the transition matrices to have a larger diagonal (stay in the same state)?

---

> ### Author Response · Authors · 2023-11-21
>
> Thanks for your further questions.
>
> 1. Given the subpopulations problem is not the primary goal of our models, we agree with you that the relative performance might not be as good as the example in the main content, as you pointed out the spurious/noisy weights should not have appeared at the off-diagonal blocks. As we mentioned before, we first reduced the subpopulations problem and then used our OHG to solve it. Therefore, OHG doesn't know that the off-diagonal blocks should be all 0s. There are two possible solutions:
>     * Increase the data size, so that OHG could learn better from the more sufficient dataset that the off-diagonal blocks could be 0s. We doubled the data size and the results became slightly better (but not visually significant). Those off-diagonal blocks are still not completely silenced.
>     * If we know the configurations of the subpopulations, we can certainly use a regularization term to suppress the off-diagonal blocks, or even silence them in a hard style (e.g., through a mask). Then, we can obtain weight matrices with less noisy elements on those off-diagonal blocks (Fig. 15 in Appendix A.6).
> 2. Thanks. Please check the updated Fig. 10 in Appendix A.4.3. The lowest range of BIC $S\in\\{3,4,5\\}$ roughly matches the turning point of the log-likelihood plot.
> 3. Adding a prior to strengthen the diagonal of the transition matrix is an intuitive way of suppressing the fast state switches in HG. This is equivalent to applying an L2 regularization term on the off-diagonal elements in the transition matrix. For some applications, it might be helpful. On the PFC-6 dataset, however, Fig. 18 in Appendix A.9 shows that when the number of state switches is suppressed, the inferred states are still meaningless. Particularly, most switches happen within a short duration, which looks like nothing but noisy state switches. Only one major state governs the whole trial. Besides, Fig. 8 in Appendix A.4.1 shows that both GHG and OHG don't have such a fast switches phenomenon. This might further imply that a global structural prior plays an important role in HMM-GLM-based models.

---

### Author Response · Authors · 2023-11-16
**Global reply**

Dear all reviewers and ACs,

Thank you for dedicating your time to reviewing our paper and providing valuable feedback.
* We have incorporated new experimental results into the Appendix in our latest revision, which we encourage you to examine. New results like Fig. 9, 10, 11, and 14 are particularly interesting.
* As suggested by Reviewers zShZ and G7kx, we reorganized Section 2 to make the methods introduced logically better.
* Additionally, we have re-plotted Fig. 4, 5, 7, 8, and 9. Notably, the state predictions are now presented in the form of segment plots, enhancing readability and providing a more elegant and comprehensible visualization.

We welcome everyone to explore these updates. Thanks again for your engagement!

---

### Author Response · Authors · 2023-11-21
**Hope to hear your feedbacks**

Dear reviewers,

According to all your valuable questions and suggestions, we have iterated several versions of our paper. There are a lot of changes in the main content and new results and figures in the Appendix in our latest revision. Hopefully, these new materials in our latest revision could solve most of your questions and concerns.

Since it is already very close to the discussion deadline, if you still have further questions about our work, please don't hesitate to ask us as soon as possible. We will try our best to answer them before the discussion deadline.

Thanks again for your time and valuable comments.

---

### Meta-Review · Area_Chair_mdMT · 2023-12-11

**Metareview:**

The paper investigates an extension of GLM based approaches for spike count
data in electrophysiology signals to capture time-varying functional interactions.
The method is called a "HMM-GLM" approach as the latent factors governing
the temporal state dependencies is considered Markovian.

The paper is well motivated given past literature on GLM for spike data.
It also comes with some mathematical considerations given the novel
HMM latent model with time-varying interactions. It is also evaluated
on two datasets where results can be related to the behavior of the rodent
doing the task. Quantitative evaluation involves estimating the likelihood
of left-out data.

The work has received strong positive feedback from 2 reviewers, feedbacks
which are aligned with the evaluation of the AC.

**Justification For Why Not Higher Score:**

The contribution is of limited scope of the ICLR audience.

**Justification For Why Not Lower Score:**

The paper is well motivated, well illustrated with real data and comes with some mathematical content on the statistical inference.

---

### Decision · Program_Chairs · 2024-01-16

Accept (poster)